# Dual Process Learning: Controlling the Use of In-Context vs. In-Weights Strategies with Weight Forgetting

**Suraj Anand**    **Michael A. Lepori**    **Jack Merullo**    **Ellie Pavlick**

Department of Computer Science
Brown University
Correspondence to `surajk610@gmail.com`.

## Abstract

Language models have the ability to perform in-context learning (ICL), allowing them to flexibly adapt their behavior based on context. This contrasts with in-weights learning (IWL), where memorized information is encoded in model parameters after iterated observations of data. An ideal model should be able to flexibly deploy both of these abilities. Despite their apparent ability to learn in-context, language models are known to struggle when faced with unseen or rarely seen tokens (Land & Bartolo, 2024). Hence, we study **structural in-context learning**, which we define as the ability of a model to execute in-context learning on arbitrary novel tokens – so called because the model must generalize on the basis of e.g. sentence structure or task structure, rather than content encoded in token embeddings. We study structural in-context algorithms on both synthetic and naturalistic tasks using toy models, masked language models, and autoregressive language models. We find that structural ICL appears before quickly disappearing early in LM pretraining. While it has been shown that ICL can diminish during training (Singh et al., 2023), we find that prior work does not account for structural ICL. Building on Chen et al. (2024)'s active forgetting method, we introduce pretraining and finetuning methods that can modulate the preference for structural ICL and IWL. Importantly, this allows us to induce a *dual process strategy* where in-context and in-weights solutions coexist within a single model.[1]

## 1 Introduction

A fundamental trait of transformer language models (LMs) is their ability to integrate context to adjust model representations and behavior without weight updates. This ability enables emergent phenomenon such as 'in-context' learning (ICL) (Brown et al., 2020; Dong et al., 2023; Garg et al., 2023), and more generally allows models to flexibly accommodate variations in language. For instance, a model is likely to memorize that the token *green* is typically used as an adjective, yet still recognize that it is used as a noun in the sentence *The child sat on the main green* based on contextual information.

However, this flexibility breaks down on truly novel/unseen tokens. Recent research has found that models cannot successfully perform ICL when given undertrained (Land & Bartolo, 2024; Rumbelow & Watkins, 2023) or newly-introduced tokens (e.g. when adding languages to an existing model) (Chen et al., 2024). For example, otherwise-performant language models produce bizarre responses when queried to simply repeat an undertrained token (Rumbelow & Watkins, 2023). Notably, this task does not require any semantic content to be encoded within the embedding of the token of interest, and so one might expect a model to implement a solution that is robust to undertraining.

In light of this, we distinguish two types of strategies that a language model might implement when faced with a task presented in context: **conditional ICL** refers to strategies that are sensitive to

---

[1] We release code at `https://github.com/surajK610/dual-process-learning` for reproducibility

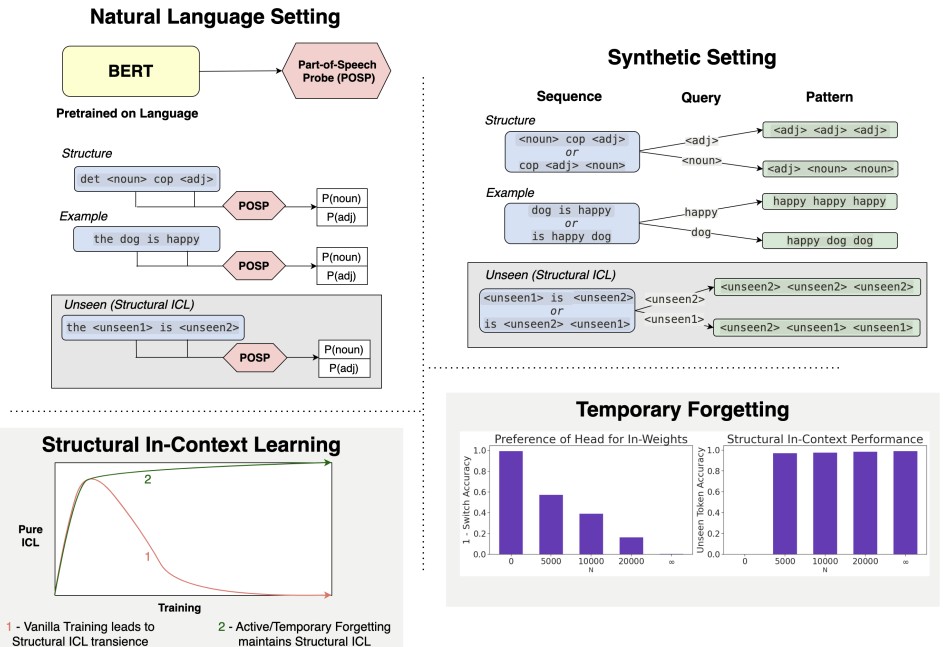

Figure 1: (Top Left) In our naturalistic setting, we train a part-of-speech probe on BERT representations of sentences from Penn Treebank 3 and evaluate it on templatic examples (Section 3). (Top Right) In our synthetic setting, we train a small masked language model (MLM) on sequences where the expected response is determined based on the part-of-speech of the query token (Section 4). (Bottom Left) An idealization of two main findings: (1) structural ICL is transient (i.e. decays over training) in both naturalistic and synthetic settings, and (2) Active/temporary forgetting maintains structural ICL in the synthetic setting. (Bottom Right) Temporary forgetting induces structural ICL when applied for $N > 0$ steps, enabling generalization to unseen random tokens. In-weights preference is coarsely controllable by varying temporary forgetting parameter $N$.

the semantic content of all tokens, whereas **structural ICL** refers to strategies that are invariant to the information (or lack thereof) encoded in the embedding weights of at least one token. Refer to Section 2 for a more precise definition. While not all tasks permit structural ICL strategies, several fundamental syntactic (Section 3) and logical (Section 7) tasks do.

A recent influential line of work has studied the development of transformers throughout training through the lens of ICL vs. in-weights learning (IWL). This has resulted in several notable findings, including (1) that models often adopt *either* ICL or IWL strategies, unless the data distribution has specific, language-like properties (Chan et al., 2022b), (2) that ICL is *transient*, disappearing as the models become overtrained (Singh et al., 2023), and (3) that $L_2$-regularization mitigates ICL transience, but instead leads to IWL transience (Singh et al., 2023). We expand upon this framework to study the relationship between conditional ICL, structural ICL, and IWL. Moreover, we aim to translate insights from these studies into actionable techniques to encourage models to flexibly deploy *both* IWL and structural ICL strategies. We refer to this capability as **dual process learning** in loose analogy to Dual Process Theory (Kahneman, 2011), as IWL implements automatic, memorized operations for IID settings (à la System 1) and structural ICL enables flexible, context-sensitive operations for out-of-distribution settings (à la System 2).

In the present study, we find that structural ICL is *also* transient. However, while regularization provides a path to persistence for conditional ICL (Singh et al., 2023), it does not for structural ICL. Therefore, we propose an extension to active forgetting – a recent weight resetting technique introduced by Chen et al. (2024) to help augment models with new tokens – to render structural ICL persistent. Our modification allows us to coarsely control the strategies that the model adopts during pretraining, enabling us to induce dual process learning: (structural) ICL for rare and unseen tokens and IWL for common tokens. Finally, we demonstrate a proof-of-concept fine-tuning strategy to induce dual process learning in pretrained causal language models.

In summary, our main contributions are:

- We define and study the concept of **structural ICL** in both large models and toy models. We discover that both masked and autoregressive LMs exhibit a (limited) form of structural in-context learning that emerges early in training, but this ability quickly vanishes.

- We show that active forgetting (Chen et al., 2024) maintains structural ICL in models.

- We introduce **temporary forgetting**, which enables one to control how much a model relies on in-weights vs. in-context strategies. We find that temporary forgetting enables us to induce dual process learning under a variety of data distributions, where our model uses an in-weights strategy for frequently tokens and a (structural) in-context solution for rarely seen tokens.

- We introduce **probabilistic temporary forgetting**, which enables one to induce structural ICL in a pretrained causal language model. We demonstrate a proof-of-concept by fine-tuning GPT-2 and demonstrating structural ICL in a simple logical reasoning task.

## 2    DEFINITIONS

**In-Context vs. In-Weights Learning**    We follow Reddy (2023), which defines in-weights learning (IWL) to be "query-response relationships encoded in the weights of the network" while in-context learning (ICL) emerges due to "common structural element[s]" and "can be exploited to perform zero-shot learning on novel tasks that share this structure." Notably, word embeddings are purely in-weight representations, which are enriched with contextual information by attention layers.

We formulate our ICL prediction tasks as $\mathbb{P}_{\mathbf{M}}(y \mid \mathbf{p}_{1:n}; \mathbf{z}_{1:n})$ where $y$ are the label(s), $\mathbf{p}_{1:n}$ is the set of positional embeddings, $\mathbf{z}_{1:n}$ is the set of word embeddings for a sequence of length $n$, and $\mathbf{M}$ refers to the parameters of a language model.

**Structural vs. Conditional ICL**    We define structural ICL precisely via an empirical test: a model exhibits structural ICL if it can complete a task that is presented in context in a way that is invariant to the content of one or more embeddings. For one or more word embeddings at specified position(s) $i \in I$, we replace $\mathbf{z}_i \xrightarrow{\text{replace}} \mathbf{z}_{\text{random}}$. This removes the in-weight information contained within the word embedding and forces the model to rely on on contextual information and structural analogy. We state that a model can perform conditional ICL when it succeeds on prediction task when $\mathbf{z}_{1:n}$ remains unmodified. This is the standard ICL setting studied in the literature (Chan et al., 2022b; Singh et al., 2023; Garg et al., 2023; Akyürek et al., 2024). Note that a model that exhibits conditional ICL will not necessarily exhibit structural ICL.

**Head vs. Tail**    In skewed token distributions, we refer to the most frequently occurring tokens (typically $\approx 10\%$) as the **head** of the distribution and the least frequently occurring tokens (typically $\approx 10\%$) as the **tail**. As token distributions increase in skew, tail tokens are seen less frequently. This dichotomy relates to our analysis of structural ICL because tail tokens are an interpolation between fully-trained tokens and random tokens. By accommodating random tokens through structural ICL, we can also recover performance on infrequent tail tokens.

## 3    (STRUCTURAL) IN-CONTEXT LEARNING IS TRANSIENT

Recent work has discovered that conditional ICL capabilities slowly degrade over the course of long training in a synthetic setting (Singh et al., 2023). In this section, we study the transience of conditional and structural ICL over the course of training in a naturalistic setting using BERT-style models (Devlin et al., 2019). Using a syntax probing task, we find that structural ICL rapidly degrades to completely random performance after fairly few training steps, while conditional ICL abilities degrade during a significantly longer timescale. To perform this analysis, we use intermediate checkpoints released from the MultiBERTs (Sellam et al., 2021), averaging all of our results across seeds 0, 1, and 2. We calculate error bars in Figure 2 as $\pm 1$ standard error of the mean (SEM).

## 3.1 TASK

Determining the part of speech (POS) of each word in a sentence is a fundamental step toward understanding that sentence. We identify two strategies that a model might employ to determine the POS of a token: (1) an in-weights strategy, where the model explicitly memorizes the POS of a token in its weights, and (2) an in-context strategy, where the model infers the POS of a given token from context information. We created several templatic evaluation datasets in order to tease these two strategies apart. Each dataset contains sentences that obey the template: The `<noun>` is `<adj>` (e.g. *The dog is happy*).

Our evaluation datasets are defined as follows:

1. **Head (Tail)**: Contains sentences where filler tokens are sampled from the most (least) frequent 1500 nouns and most (least) frequent 1500 adjectives in the training set of Penn Treebank 3 (PTB-3) (Marcus et al., 1993). We sample 1500 unique words because this comprises $\approx 10\%$ of all unique nouns in PTB-3.
2. **Head (Tail) Switch**: Contains sentences where tokens are sampled as in the "Head" ("Tail") dataset, but where noun tokens fill the `<adj>` slot and adjective tokens fill the `<noun>` slot (e.g., *The happy is dog*). These datasets put the IWL strategy and ICL strategy into conflict.
3. **Random Token**: Contains sentences where the `<noun>` and `<adj>` slots are filled by randomly-initialized embeddings. This dataset evaluates structural ICL performance. This dataset appeals to the intuition that it should be possible to infer the POS of nonce tokens in sentences like "the gluck is wug."

For each layer and MultiBERT checkpoint, we train a separate binary POS probe on representations of nouns and adjectives from sentences in the training set of PTB-3 (Marcus et al., 1993). We then evaluate these trained probes on our evaluation datasets in order to understand the strategies that models employ to determine POS at various points through training. For multi-token words, we average representations across tokens (See Appendix A for additional details). Note that the MultiBERTs are trained following Devlin et al. (2019) on a combination of BookCorpus (Zhu et al., 2015) and English Wikipedia collected by Turc et al. (2019). As such, the distribution of the training data is fixed, and our experiments are constrained to the natural distribution of language. As BookCorpus does not have POS tags readily accessible, we employ PTB-3 to estimate the noun and adjective distribution of the training data. We classify a word as either nouns (adjectives) if that word appears as a noun (adjective) over 80% of the time. See Figure 1 (Top Left) for more details.

## 3.2 TRAINING DYNAMICS

We examine (1) structural in-context learning and (2) the tradeoff between in-context and in-weight strategies over the course of training.

**Structural ICL** We find that the MultiBERTs are able to perform structural ICL early in training, but that this capability is transient. We measure structual ICL by evaluating pretrained probe performance on the **Random Token** evaluation dataset. If a model determines POS using an in-context strategy that is invariant to the content of the probed token, then it should succeed at inferring the POS of the random tokens inserted in both slots. In Figure 2 (Left), we present accuracy on the **Random Token** dataset using a probe trained on representations from Layer 7, as this layer achieves the highest probing validation performance on PTB-3 (See Appendix B for results across all layers). We find a clear signature of structural ICL transience: probe performance on random tokens spikes early in MultiBERT training before dropping to chance by the end of training. These results suggest that there is an inductive bias toward structural ICL that diminishes as information is encoded in the model weights.[2] As structural ICL confers the ability to generalize to rare and new tokens, this finding raises questions about how we can train models that maintain this ability throughout training.

**In-Context vs. In-Weights Strategies** Much like Singh et al. (2023), we observe that conditional ICL strategies decay over training as information is encoded in model weights (e.g., token embeddings). To approximate how much a model relies on contextual information to infer the POS of tokens, we consider the difference in performance between probes trained on Layer 0 (the embedding layer) and probes trained on Layer 7 on the **Head** and **Tail** evaluation datasets. Layer 0 must rely only

---

[2]We also observe structural ICL transience in Pythia-1.4B (See Appendix C)

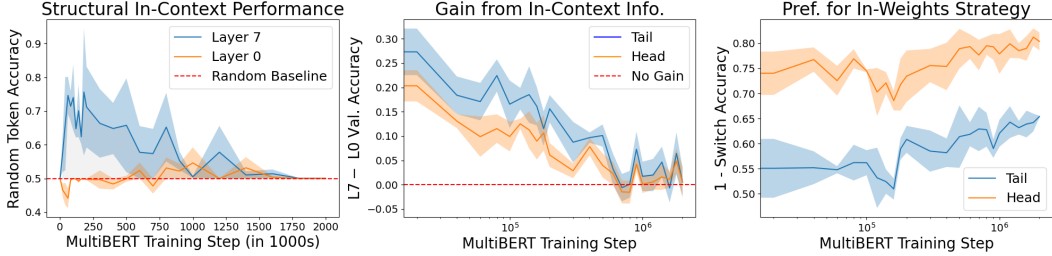

Figure 2: (Left) Structural ICL is transient, as **Random Token** accuracy first peaks and then decays. (Middle) We investigate the benefit of contextualization over memorization in **Head** and **Tail** datasets by examining the difference in Layer 7 Accuracy (where both in-context and in-weights strategies are possible) and Layer 0 Accuracy (where only an in-weights strategy is possible). These differences become negligible after sufficient training. (Right) Using the **Head Switch** and **Tail Switch** datasets, we find that models begin to encode POS using an IWL strategy over time. Note that the x-axis begins at training step 20,000 for (Middle) and (Right).

on information encoded in the embedding matrix, as there is no in-context information available; in contrast, Layer 7 can use contextualization to achieve higher performance (Tenney et al., 2019; Hewitt et al., 2021). We find that the benefit of contextualization fades for both **Head** and **Tail** datasets, but dissipates more quickly for the head of the distribution than the tail (See Figure 2, Middle). We hypothesize that this occurs because there are far more gradient updates to head token embeddings.[3] Concurrently, we measure Layer 7 probe performance on the **Head Switch** and **Tail Switch** datasets. We observe that the model shifts from an in-context to an in-weights strategy, preferring to infer POS from token identity, rather than token position (See Figure 2, Right). In other words, models become more reliant on in-weights strategies and less reliant on in-context strategies over the course of training. This finding aligns with Singh et al. (2023).

## 4 DATA DISTRIBUTION IMPACTS IN-CONTEXT LEARNING

We develop a synthetic masked language modeling task to characterize how data distributional parameters affect structural ICL, conditional ICL, and IWL. Our synthetic task requires the model to determine which of two classes a word belongs to. A token's class may be inferred from contextual information or memorized in the embedding layer. This task was crafted as a simplified version of the naturalistic task investigated in Section 3.

Our vocabulary contains tokens that represent nouns, adjectives, and a copula (i.e., *is*). Each input sample is created by selecting (1) a `sequence` $S$, (3) a `query` $Q$, (3) two filler tokens $x_{noun}$, $x_{adj}$. The `query` uniquely determines a `response pattern` $P$. Our MLM is trained to predict $\mathbb{P}(P_i|S, Q)$ for all $i \in \{0, \ldots, |P| - 1\}$ (i.e. the probability of each pattern token). The `sequence` and `pattern` are designed so that no exceedingly simple heuristic can solve this task. Specifically, sentence templates are defined using the following elements:

- **`sequence` $S$**: Either `<noun> <copula> <adj>` or `<copula> <adj> <noun>`.
- **`query` $Q$**: Either `<noun>` or `<adj>`.
- **`response pattern` $P$**: Either `<adj> <noun> <noun>` if the query is `<noun>` or `<adj> <adj> <adj>` if the query is `<adj>`.

These templates are populated by $x_{noun}$ filling the `<noun>` slots and $x_{adj}$ filling the `<adj>` slots. This task is designed such that the model must make a POS classification on the query token, and then perform an additional operations conditioned on that classification (i.e., copying specific token identities in a specific order). See Figure 1 (Top Right) and Appendix I for examples.

We parameterize the task with vocabulary size $v$, the sampling distribution skew for noun/adjective fillers $\alpha$ (where we select $x_{noun}, x_{adj} \sim \text{Zipf}(\alpha)$), and the ambiguity of token POS $\varepsilon$. The ambiguity

---

[3]We observe that performance gain due to the model's use of in-context information decreases across a wide range of syntactic phenomena as embeddings are enriched during training. We term this the "Pushdown Phenomenon" and explore it more thoroughly in Appendix H.

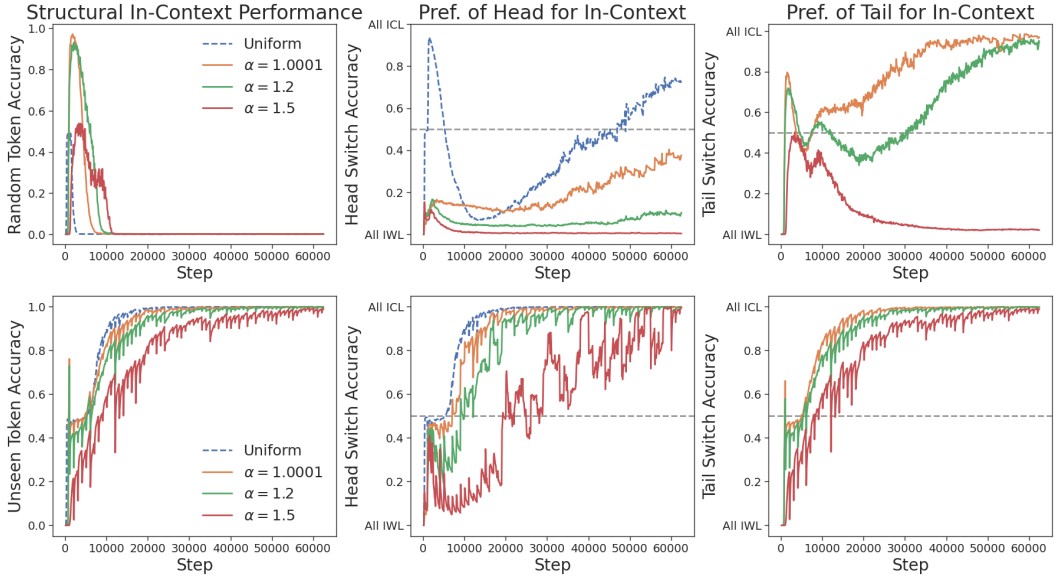

Figure 3: Comparative analysis of in-context learning performance across training methodologies and data distributions. (Top) In-context performance by distribution with **vanilla training**; (Bottom) In-context performance by distribution with **active forgetting**. The parameters used are $v = 10000, \varepsilon = 0.10$. Note that the Uniform distribution does not have a head or a tail, and we present results in the head graphs. (Top Left) Vanilla training results in structural ICL transience across all distributions. (Top Middle, Top Right) Conditional ICL is asymptotically nonzero for most distributions, unless they are highly skewed (i.e., $\alpha = 1.5$). (Top Middle) However, IWL is often preferred for head tokens and (Top Right) conditional ICL is preferred for tail tokens. (Bottom Row) In contrast, active forgetting preserves structural ICL and removes all preference for IWL across distributions and datasets. **Note:** The y-axis in the bottom left is relabelled "Unseen Token Accuracy" to emphasize that the random token evaluation dataset does not contain any random embeddings seen during active forgetting.

parameter determines the percentage of filler tokens can fill both `<noun>` and `<adj>` slots, and is inspired by the ambiguity of POS found in natural language. For our primary experiments, we fix $\varepsilon = 0.10$.[4] We investigate how training dynamics change as the skew changes, and additionally compare to uniform sampling distribution.

In this task, an ICL strategy to infer the POS of the `query` token may achieve perfect accuracy by utilizing in-context information (e.g. a *copula* is always followed first by an adjective, then a noun). In contrast, an IWL strategy may achieve an accuracy of $(1 - \varepsilon/2)$ at most due to ambiguous tokens. Thus ambiguity provides mild pressure to develop an ICL strategy. In order to make fair comparisons, we only evaluate our models on the subset of tokens that are not ambiguous; thus, both an ICL and IWL solution could achieve perfect accuracy.

Our task is formatted in a cloze-style where each token in the pattern is masked. We employ a BERT-style MLM (Devlin et al., 2019) to predict the identities of these masked tokens, with hyperparameters described in Appendix J. Our models achieve near-perfect validation accuracy after $< 60,000$ steps in all experimental settings.

In addition to performance on a randomly selected validation set, we create datasets to evaluate the model's preferred strategy throughout training, similar to Section 3. All examples in these datasets contain novel $\{x_{noun}, x_{adj}\}$ combinations. Much like our naturalistic setting in Section 3.1, we create **Head**, **Tail**, **Head Switch**, **Tail Switch**, and **Random Token** datasets. In this setting, our head and tail datasets use the top and bottom 10% of the token distribution by count, respectively.

---

[4]Interestingly, we find that $\varepsilon$ must be greater than zero for an in-context solution to emerge at all.

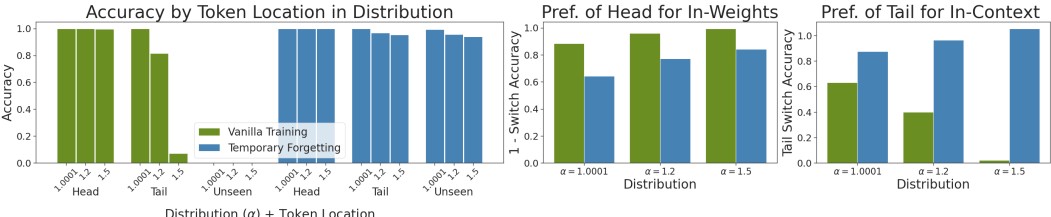

Figure 4: (Left) Temporary forgetting achieves near perfect unseen random token performance across distributions, indicating structural ICL. (Left, Green) Vanilla training on skewed distributions renders tail token performance poor; (Left, Blue) In contrast, tail token performance is almost perfect after temporary forgetting. (Right) Temporary forgetting can maintain a preference for IWL for the head of the distribution while maintaining a preference for ICL for the tail of the distribution i.e., temporary forgetting induces dual processes learning. Parameters used are $v = 10000, \varepsilon = 0.10$ and optimal hyperparameters $k, N$ are found using a grid search.

## 4.1 TRAINING DYNAMICS

**Transience of Structural ICL**   We reproduce our results from the naturalistic setting presented in Section 3: structural in-context strategies emerge quickly, but are transient. This is shown by the model's performance on the **Random Token** dataset over the course of training, which peaks early and then quickly degrades (See Figure 3, Top Left). This trend holds across all tested distributions. Thus, both synthetic and naturalistic training settings result in structual ICL transience, as hypothesized in Figure 1 (Bottom Left). Critically, we find that models retain conditional ICL strategies, even after structural ICL performance degrades. Across most data distributions, performance on both the **Head Switch** and **Tail Switch** datasets reveal a nonzero reliance on conditional ICL strategies, even while **Random Token** accuracy remains at zero (See Figure 3 (Top Middle, Top Right)).

**Structural ICL has Practical Importance** In highly skewed distributions (e.g. Zipf $\alpha \geq 1.5$) where tail tokens are very rare and head tokens are very common, the disappearance of structural ICL eventually precipitates a total loss of ICL abilities (Figure 3 Top, Red Line). We find that common tokens are memorized, resulting in high overall performance. However, the model completely fails when presented with tail tokens (See Figure 4, Left, $\alpha = \{1.2, 1.5\}$). Even when conditional ICL strategies remain after training in less-skewed distributions, the least-frequent subset of tail tokens result in poor performance. Inducing structural ICL would recover performance on these undertrained tokens.

**In-Context Learning conflicts with In-Weights Learning**   We analyze how the skew of the training distribution applies pressure toward adopting an IWL or ICL strategy. We find that increasing the skew of a distribution increases the pressure toward an IWL strategy for the head of the distribution, and increases the pressure toward an ICL strategy for the tail of the distribution. Furthermore, training distributions with a Uniform sampling distribution show a comparatively higher conditional ICL preference (and thus lower IWL preference) than any Zipfian sampling distribution (See Figure 3, Top Middle). We explore how to mitigate this competition in Section 6.[5]

## 5 MAINTAINING STRUCTURAL ICL WITH ACTIVE FORGETTING

In Sections 3 and 4, we have demonstrated structural ICL is transient across models and tasks. In an effort to promote the persistence of structural ICL, we utilize *active forgetting* (Chen et al., 2024). Henceforth, we refer to the standard training procedure as *vanilla training*.

**Active Forgetting**   When training a model using active forgetting, we re-initialize the embedding matrix every $k$ steps during training. The intuition behind this is that the model *must* employ in-context strategies to achieve high accuracy, as each embedding is no longer guaranteed to encode

---

[5]Additional experiments exploring the effect of ambiguity are located in Appendix L and the effect of vocabulary size are located in Appendix M.

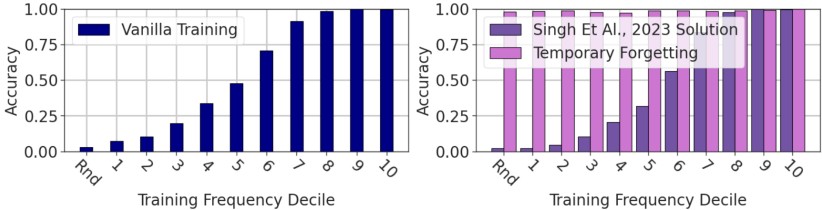

Figure 5: Performance by token decile and on randomly initialized embeddings (Rnd). (Left) With vanilla training on a skewed distribution (Zipfian $\alpha = 1.5$), low decile tokens show poor performance. However, overall performance remains good because these tokens are rare. (Right) Temporary forgetting induces structural ICL to recover performance on tail, undertrained, and unseen tokens compared with Singh et al. (2023)'s $L_2$-regularization procedure, which was proposed to preserve conditional ICL.

in-weight information. This renders unseen/undertrained embeddings in-distribution, whereas they were previously out-of-distribution. We further explore this effect in Appendix N.

**Results** We test $k = 100, 1000, 5000$ and settle on $k = 1000$ after preliminary exploration. Training our models with active forgetting asymptotically promotes structural ICL across all tested skews. We evaluate this using the **Random Token** dataset, and find near-perfect performance after sufficient training on all distributions (See Figure 3, Bottom Left). Given random embeddings to fill the `<noun>` and `<adj>` slots, the model can now (1) derive the POS of these tokens by ICL and (2) output novel labels corresponding to the identity of these embeddings in the desired pattern.[6]

As the skew of the distribution of nouns and adjectives increases, there is greater pressure to memorize the head of the distribution (as these tokens are observed more frequently). Thus, it takes longer for the model to exhibit a preference towards in-context solutions for head tokens (e.g., almost 60,000 steps for the $\alpha = 1.5$ setting) and there is a much larger dip in performance after every instance of forgetting the embedding matrix. However, we find that our active forgetting results generally match our idealized result from Figure 1 (Bottom Left).

## 6 DUAL PROCESS LEARNING WITH TEMPORARY FORGETTING

While active forgetting successfully induces a structural ICL strategy, our model loses the ability to memorize information in its embeddings. This is detrimental in a variety of cases, such as when in-context information is insufficient to generate an appropriate response. An optimal model would encode a *dual process strategy*: maintaining a structural ICL solution while also memorizing useful linguistic properties.

**Temporary Forgetting** We modify the paradigm of active forgetting in order to induce a bias for structural in-context strategies for the tail of the distribution while preserving in-weights strategies for frequently-observed tokens. We introduce **temporary forgetting**, where we perform active forgetting every $k$ steps for the first $N$ steps ($N >> k$) of training. After this point, we allow the embedding matrix to train normally. As a baseline, we compare to Singh et al. (2023)'s solution to conditional ICL transience, $L_2$ regularization. Crucially, we wish to understand whether $L_2$ regularization helps maintain *structural* ICL, which was not tested in the original work.

**Results** We study a highly skewed distribution, with parameters $v = 10000, \varepsilon = 0.10, \alpha = 1.5$. We find that by varying $N$, we can vary the model's dependence on in-weights information for frequently seen tokens while maintaining structural ICL performance (See Figure 1, Bottom Right). At the extremes, setting $N$ to be very large mimics the behavior of active forgetting and setting $N$ to be small only *sometimes* maintains structural ICL performance. We can control the preference for IWL versus ICL on observed tokens by modifying $N$.

Thus, temporary forgetting enables a model to successfully encode two distinct strategies for the same task. We can now induce this behavior for any distribution $\alpha \geq 1.0$ (See Figure 4, Right),

---

[6]This is possible because the embedding and unembedding matrices being tied.

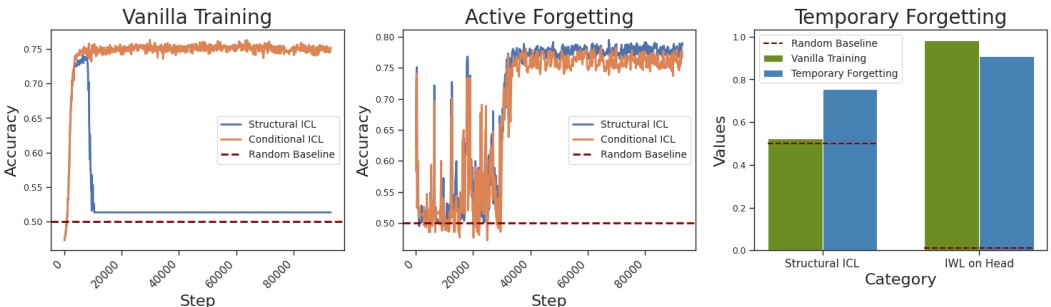

(a) Results from our replication study on the Chan et al. (2022b) task with an autoregressive transformer. (Left) With vanilla training, structural ICL is transient while conditional ICL is persistent. (Middle) Training with active forgetting preserves structural ICL. (Right) When training our model to a skewed token distribution (Zipfian $\alpha = 3$), vanilla training results in memorization of head tokens and chance performance on unseen tokens; in contrast, temporary forgetting evokes a dual process, which significantly improves structural ICL performance while preserving IWL on common tokens. Further task and experiment details in Appendix E.

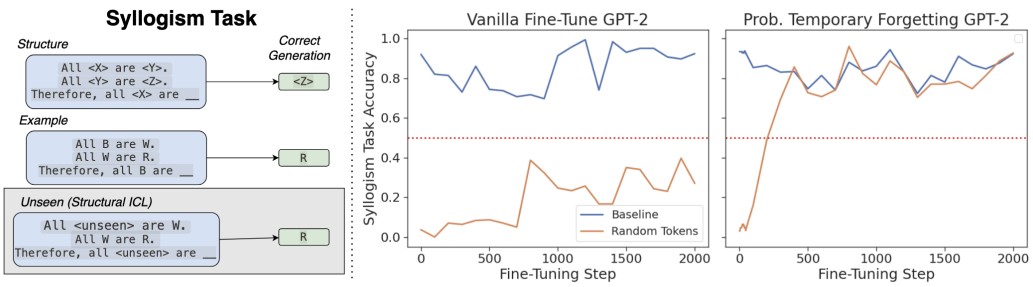

(b) Probabilistic temporary forgetting enables GPT-2 to perform structural ICL. (Left) A diagram representing the syllogism task. (Middle) Vanilla fine-tuning GPT-2 large on Wikitext fails to confer structural ICL, resulting in poor Random (Unseen) Token Performance. (Right) In contrast, we find that probabilistic temporary forgetting successfully confers structural ICL, drastically improving Random Token Performance.

Figure 6: Autoregressive Transformer Structural ICL Experiments

while also inducing structural ICL behavior on *all* distributions we test (See Figure 4, Left).[7] In contrast, we find that the strategy employed by (Singh et al., 2023) does *not* eliminate structural ICL transience: undertrained and random tokens result in poor performance, as seen in Figure 5. In summary, temporary forgetting significantly enhances our ability to balance between in-context and in-weights strategies, overcoming inherent biases in naturally occurring data. After a critical number of training steps, we can stop the forgetting mechanism and retain structural ICL.

# 7 STRUCTURAL ICL IN AUTOREGRESSIVE TRANSFORMERS

**Replication using Chan et al. (2022b) Task** In this section, we replicate our main findings using an autoregressive transformer on a task similar to Chan et al. (2022b). We modify the task presented in Chan et al. (2022b) to enable us to examine structural ICL (See Appendix E).[8] We find that the phenomena described in Sections 3, 4.1, 5, and 6 all extend to this new task. In particular, Figure 6a (Left) demonstrates the transience of structural ICL, even when conditional ICL persists. Figure 6a (Middle) demonstrates that active forgetting preserves structural ICL. Finally, Figure 6a (Right) demonstrates that temporary forgetting induces a dual process strategy, where structural ICL is maintained *and* IWL is deployed on the head of a skewed distribution.

---

[7]Distributions where $\alpha \leq 1.0$ would likely only rely on an in-context strategy.

[8]This modification ensures the tail distribution of these tokens are undertrained/untrained and thereby resemble the "glitch tokens" of Rumbelow & Watkins (2023).

**Probabilistic Temporary Forgetting Induces Structural ICL in GPT-2**   Thus far, our interventions have focused exclusively on pretraining to induce structural in-context learning (ICL). In this section, we expand this approach to explore whether fine-tuning can also facilitate dual process learning. As a proof-of-concept, we introduce **probabilistic temporary forgetting**. Our methodology involves fine-tuning a GPT-2 model using a causal language modeling objective, with one key modification: during each fine-tuning step, we replace approximately 10% of tokens in the batch with randomly-initialized embeddings. Importantly, after each gradient update, we restore the embedding matrix to its original pretrained values. For additional details, please refer to Appendix C.

We designed a simple syllogism task inspired by Lampinen et al. (2024) and Kim et al. (2024) (Figure 6b, Left). Notably, we include a condition where the subject term in the first premise and conclusion of the syllogism is replaced by a random embedding. As we use an unconditionally valid syllogism, this should not interfere with the model's ability to perform logical inference. We fine-tune GPT-2 large (Radford et al., 2019) on Wikitext (Merity et al., 2016) sentences for 2000 steps using both (1) vanilla training and (2) probabilistic temporary forgetting.

The results reveal a striking contrast between methods. With standard fine-tuning, the model demonstrates persistently low accuracy on our syllogism task when tested with unseen tokens (Figure 6b, Middle). In contrast, when fine-tuned using probabilistic temporary forgetting, the model shows substantial improvement in handling random (unseen) tokens while maintaining comparable performance on baseline conditions (Figure 6b, Right).

## 8 DISCUSSION

**Related Work**   As discussed throughout the work, the present study is intimately related to the burgeoning literature examining the trade-offs between in-context and in-weights learning (Chan et al., 2022b;a; Reddy, 2023; Raparthy et al., 2023; Fu et al., 2024). Additionally, this work connects to a vast literature on *forgetting* in neural networks. Most prior work on forgetting characterizes this phenomenon as undesirable (Kemker et al., 2017; Kirkpatrick et al., 2017; McCloskey & Cohen, 1989; Ratcliff, 1990). However, some work has shown that *intentional* forgetting (via resetting a subset of parameters) may be beneficial in certain contexts. On computer vision tasks, forgetting has been shown to help with generalization and sample efficiency (Alabdulmohsin et al., 2021; Taha et al., 2021; Ramkumar et al., 2023). Additionally, Zhou et al. (2022) show that a *forget-and-relearn* paradigm helps shape the learning trajectory of neural networks. Our method of forgetting embeddings is directly inspired by Chen et al. (2024), which shows that forgetting during pretraining boosts linguistic plasticity for multilingual learning.

**Conclusion**   The ability to flexibly deploy in-context and in-weights algorithms has been described as an "important and useful [behavior] for a model," as it enables models to both memorize information about commonly-seen inputs and generalize to new inputs Chan et al. (2022b). However, it has proven difficult to ensure that models reliably acquire both forms of processing. This has led prior work to celebrate the ability to maintain dual strategies even for a limited set of distributions and suggest interventions such as "engineer[ing] data distributions to evoke this behavior in models" (Chan et al., 2022b). In contrast, the present work demonstrates a method for engendering dual process learning across a range of distributions. Additionally, we extend our analysis to structural in-context learning.

In summary, we find that structural ICL is transient in LMs, as they initially learn to generalize to unseen tokens, before losing this ability. We find that active forgetting recovers structural ICL, at the expense of IWL. We introduce temporary forgetting and probabilistic temporary forgetting to induce dual process learning, enabling models to leverage IWL for common tokens and structural ICL for rare or unseen tokens—this approach holds across various distributions. These strategies may prove particularly valuable for training models in domains characterized by highly skewed distributions.

This study opens several promising directions for future work. A critical next step involves determining whether temporary forgetting can be effectively incorporated into large-scale pretraining curricula, which would establish the method's broader impact. Additionally, as we provide only a proof-of-concept for probabilistic temporary forgetting, more comprehensive analysis of this technique is essential. From an implementation perspective, investigating whether probabilistic temporary forgetting can be integrated with parameter-efficient fine-tuning methods (Hu et al., 2022) represents an important advancement toward making this approach practical for large language models.

## ACKNOWLEDGMENTS

We would like to thank the members of the LUNAR, Serre, and LNCC laboratories at Brown University for their valuable feedback on this research. In addition, we would like to thank Vignesh Pandiarajan, Anish Anand, and Akash Anand for proofreading the manuscript.

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

## A    PROBING SETUP

We provide probing background in this section, borrowing some notation from Elazar et al. (2020).

Given a set of labeled data of points $X = x_1, \ldots x_n$ and task labels $Y = y_1, \ldots, y_n$, we analyze a model $f$ that predicts the labels $Y$ from $X : \hat{y}_i = f(x_i)$. We assume that this model is composed of two parts: (1) an encoder $h$ that transforms input $x_i$ into a learned representation vector $\mathbf{h}_{x_i}$ and (2) a classifier $c$ that is used for predicting $\hat{y}_i$ based on $\mathbf{h}_{x_i}$, such that $\hat{y}_i = c(h(x_i))$. We refer to $c$ as the *probe* and the model containing $h$ as the *model*.

Given this setup, we evaluate a particular model's performance across various layers and training steps for our POS task. Each encoder $h$ is associated with a specific training step and layer $h^{t,l}$. We probe the residual stream after layer $l$.

In this research, we are interested in the model's choice of strategy at a particular time step. That is, we seek to describe the change in prediction of $\hat{y}_i$ due to varying $t, l$ of encoder $h^{t,l}$. Accordingly, we fix $c$ as a single linear fully-connected layer.

## B    STRUCTURAL ICL ACROSS LAYERS

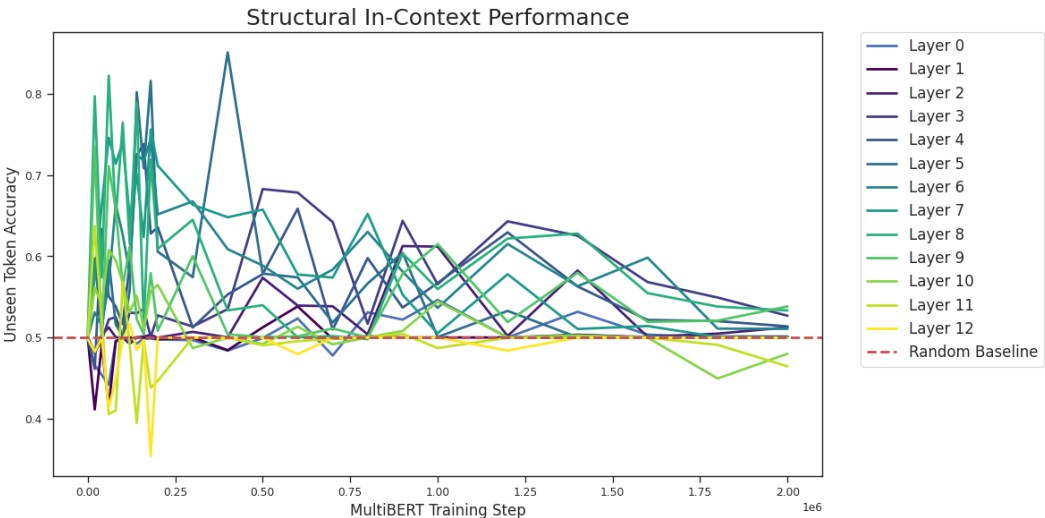

Figure 7: We find that structural ICL is transient across all layers of MultiBERTs (seeds 0, 1, 2 averaged). The middle layers show the most structural ICL during early in training, whereas very early and very late layers remain about random throughout training.

We find that structural ICL consistently approachs random levels as training progresses across layers in the MultiBERTs. This signifies that the model fully loses the ability to process unseen tokens as training continues. This is one explanation for the "glitch tokens" described in Land & Bartolo (2024), for which LMs fail to output sensible content.

## C    STRUCTURAL ICL IN GENERATIVE DECODER-ONLY LANGUAGE MODELS

### C.1    SYLLOGISM TASK

We designed a syllogism task that requires symbolic reasoning based on the context to show that (1) structural ICL is transient in a decoder-only transformer based on generation and (2) a variant of temporary forgetting can remedy structural ICL on a real-world natural langauge model.

Our task is formulated as follows: Our task requires abstract reasoning on untrained tokens in a decoder-only transformer. The model must complete the following syllogism.

All `<X>` are `<Y>`.
All `<Y>` are `<Z>`.
Therefore, all `<X>` are __

The correct answer is `<Z>`. We examine accuracy, which we define as the probability of choosing `<Z>` compared to the probability of choosing `<Y>`. We test *baseline performance* over training steps where `<X>`, `<Y>`, `<Z>` are chosen from the set of tokens representing A-Z, and we test *unseen token performance* by replacing `<X>` with an unseen token in this formulation (`<Y>`, `<Z>` are still chosen from A-Z). This task was inspired by Lampinen et al. (2024).

### C.2    STRUCTURAL ICL IS TRANSIENT IN PYTHIA 1.4B

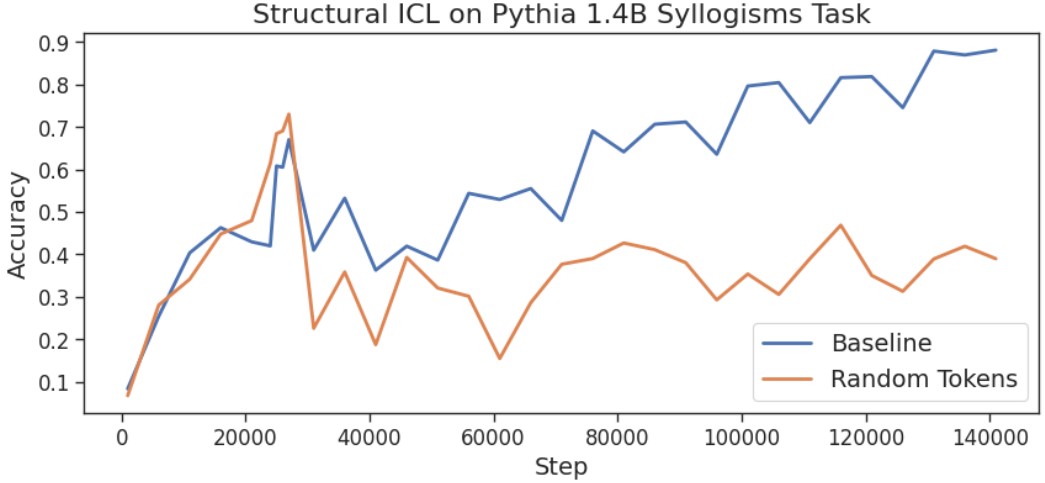

Figure 8: We find that structural ICL is transient for the decoder-only Pythia-1.4B on a syllogisms task. Performance on common tokens continues improving to near-perfect accuracy. Results averaged over three trials.

In the above syllogism task, we observe structural ICL transience in Pythia 1.4B checkpoints. We find that structural ICL consistently spikes and then approaches below random levels as training progresses across layers in the Pythia 1.4B model (Biderman et al., 2023), as shown by the random (unseen) token accuracy in Figure 8. The model loses the ability to perform syllogisms on unseen tokens as training continues. We chose the Pythia-1.4B model to show the generalizability of our finding to natural language decoder-only models. We employ publicly released training checkpoints to run our experiments (starting at step 0, then every 5000 steps starting from step 1000 until 141000).

## D    PROBABILISTIC TEMPORARY FORGETTING FIXES STRUCTURAL ICL IN GPT-2

We finetune GPT-2 large (Radford et al., 2019) on Wikitext (Merity et al., 2016) sentences taken from Wikipedia articles for 2000 steps. We use the AdamW optimizer with a learning rate of $3e-5$ and a

linear optimization schedule with 500 warmup steps. Note that unseen token syllogism performance on the pretrained GPT-2 large is even worse than on the pretrained Pythia 1.4B. To accommodate the fine-tuning setting, we use a probabilistic variant of temporary forgetting: every step, we replace tokens in the batch with $p = 0.10$ with randomly initialized embeddings. After the step, we set the embedding matrix back to its original values, hence maintaining the spirit of temporary forgetting. In this method, our pretrained embeddings remain unchanged.

After fine-tuning with probabilistic temporary forgetting on Wikipedia sentences, we find that syllogism accuracy with unseen tokens jumps from 0.02 to 0.927 while the baseline syllogism accuracy goes from 0.933 to 0.923, as seen in Figure 6b. In addition, when we fine-tune without probabilistic temporary forgetting (i.e. vanilla fine-tuning), we see that unseen token syllogism accuracy remains substantially below-random. Our probabilistic temporary forgetting rectifies structural ICL on a downstream task in a real natural language model.

# E   AUTOREGRESSIVE TRANSFORMER SYNTHETIC SETTING

To show the broadness of our structural ICL results, we also replicate our findings using a modified version of the synthetic task presented in Chan et al. (2022b).

## E.1   MODIFIED CHAN ET AL. (2022B) TASK

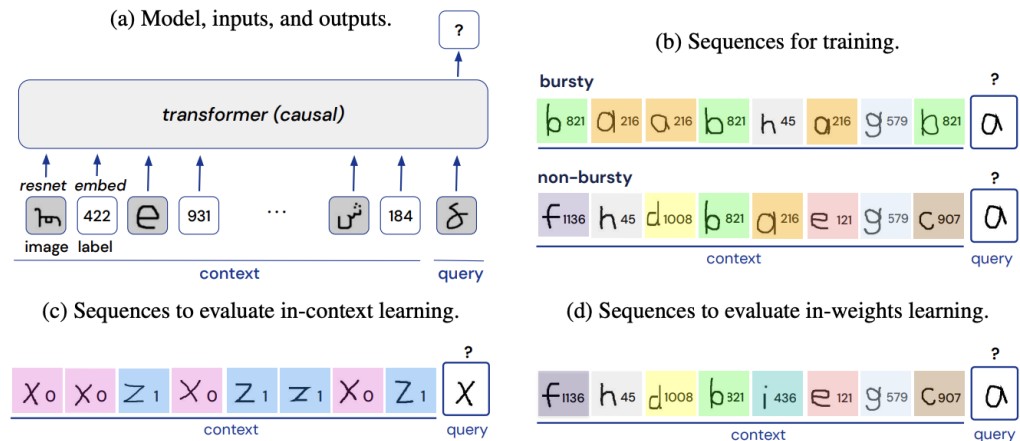

Figure 9: This is the task formulation of Chan et al. (2022b) (Replicated from Figure 1 of Chan et al. (2022b)). We use a similar task, but with token embeddings that are learned during training rather than ResNet encodings of Omniglot.

Similar to Chan et al. (2022b)'s task formulation, we have training data comprised of sequences of tokens and labels where the context is made up of the first 16 elements (8 token-label pairs), and the final element is the 'query' token. The aim of the model is to predict the correct label for the query. There are 1600 tokens, each mapped to a label. With some ambiguity probability (e.g., 0.05), a token is mapped to a different label randomly chosen from the set of labels (in order to confer ambiguity, as in Section 4). Sequences are bursty, with the query-label pair as well a different token-label pair each occurring 3 times in the context. We evaluate the trained models on three types of sequences to measure (1) structural ICL, (2) conditional ICL, and (3) IWL.

Again borrowing from Chan et al. (2022b), our context for the ICL conditions is a random ordering of two token-label pairs with 4 examples each, and the query is selected randomly from one of the two tokens. While label-pairs are fixed in training (up to the ambiguity parameter), the labels for the two tokens are randomly re-assigned to either 0 or 1 for each sequence. We calculate few-shot accuracy by considering only probabilities assigned to 0 and 1 (resulting in chance performance of 0.5). In evaluating structural ICL, we generate sequences consisting of random tokens and labels, while conditional ICL sequences consisted of tokens previously seen by the model during training. We test on tokens drawn from uniform and zipfian distributions, where experiments are with a Zipf $\alpha = 1.0001$ token sampling distribution unless otherwise specified.

To measure IWL, we considered non-bursty sequences where the query-label is not located in the context. The only way for a model to correctly predict the label is to rely on information in weights as we ensured unique, non-query token-label pairs in the context.

Note that the difference from Chan et al. (2022b)'s setup is that we use randomly initialized tokens embeddings rather than Omniglot Resnet-encoded images and our autoregressive transformer is also smaller. This enables us to test for structural ICL by replacing token identities with random vectors. Another method for us to test structural ICL could have been to use random images, but this would not maintained the analogy to undertrained/unseen "glitch tokens" in language models, unlike our current setup

### E.2 Model Description

We use a 4-layer GPT-2 architecture as our autoregressive transformer with 4 attention heads per decoder layer and an embedding size of 64 (Radford et al., 2019). To optimize, AdamW with a learning rate of $5 \times 10^{-5}$ and a linear warmup schedule with 1/10 of the total number of steps as warmup steps (Loshchilov & Hutter, 2019).

We ensure that on a validation similar to the training set, there is near-perfect performance by the completion of training.

### E.3 Vanilla Training

We find across setting that settings where ICL arises, there is structural ICL and it disappears abruptly with vanilla training. This is true for different levels of burstiness (0.8, 0.95, 1.0), different levels of ambiguity (0.05, 0.10, 0.20), and different distributions (Uniform, Zipf with $\alpha = 1.0001, 1.5, 2, 3$). In-weights learning varies based on the distribution.

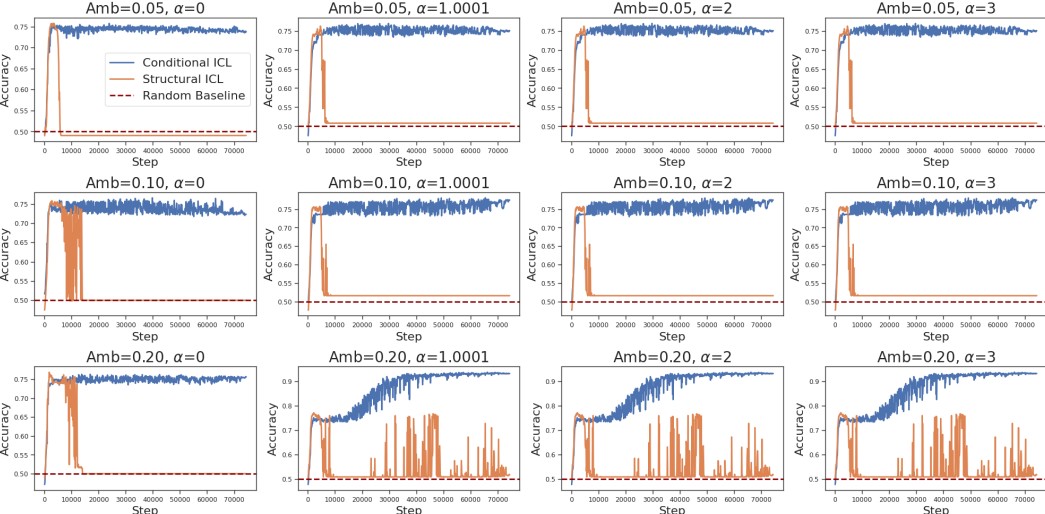

Figure 10: Structural ICL disappears while conditional ICL remains across different combinations of ambiguity and skew in our autoregressive few-shot task described in Appendix E. Interestingly, skewed distributions with high ambiguities show some variance in structural ICL accuracy after the initial disappearance.

### E.4 ACTIVE FORGETTING

Active forgetting preserves structural ICL, but completely removes any use of IWL. We see this across tested distributions (Uniform, Zipf with $\alpha = 1.0001, 2$). We use $k = 500$ because this worked well with initial experiments (although the other tested parameters of $k = 1000, 2000$ also worked almost equivalently).

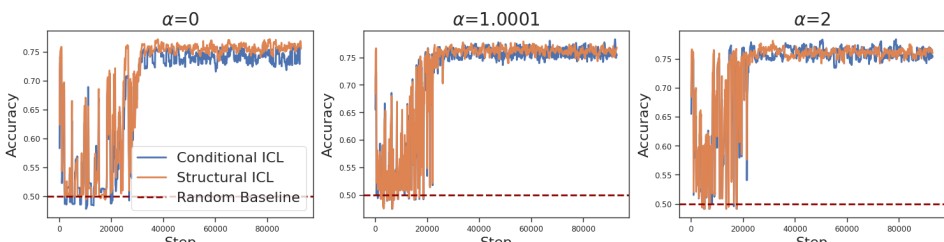

Figure 11: Active forgetting preserves structural ICL across different skews in our autoregressive few-shot task described in Appendix E. Interesting, increasing the skew seems to make active forgetting converge quicker.

### E.5 TEMPORARY FORGETTING

In our temporary forgetting setting, we use a burstiness parameter of 0.95 for experiments. We use $k = 1000, N = 8000$ because these parameters worked well in initial experiments. We did not exhaustively search over parameters. We tested whether we could evoke a dual process of ICL and IWL across distributions (Zipf with $\alpha = 1.0001, 2, 3$), as seen in Figure 12. This is in contrast to active forgetting, where we cannot learn information in-weights (Figure 13), and vanilla training, where we cannot asymptotically perform above a random baseline for structural ICL (Figure 10).

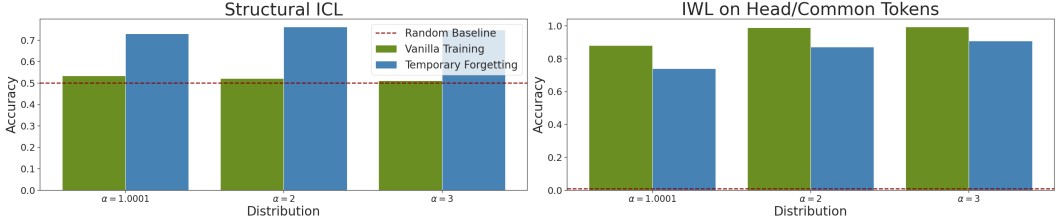

Figure 12: Temporary forgetting preserves structural ICL across different skews in our autoregressive few-shot task described in Appendix E, as opposed to vanilla training (i.e. standard training). In addition, it enables IWL for common tokens instead of completely removing it like active forgetting. It achieves about 90% the IWL use for these. Note we consider the smaller set between top 100 tokens and top 10% of the probability when choosing common tokens to evaluate IWL on.

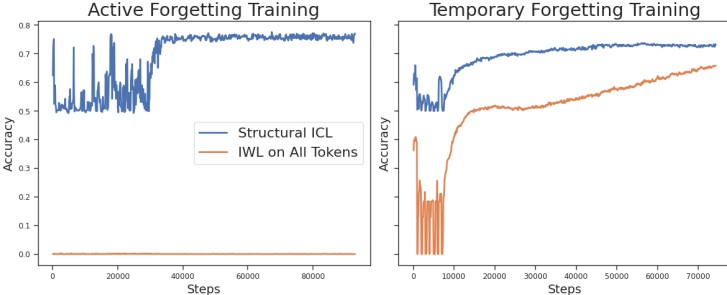

Figure 13: Temporary forgetting enables us to learn IWL while preserving structural ICL, whereas active forgetting forces only structural ICL. This is seen by the developmental accuracies in this figure (note $k = 500$ for active forgetting whereas $k = 1000, N = 8000$ for temporary forgetting).

## F   DUAL PROCESSES FOR SKEWED DISTRIBUTIONS

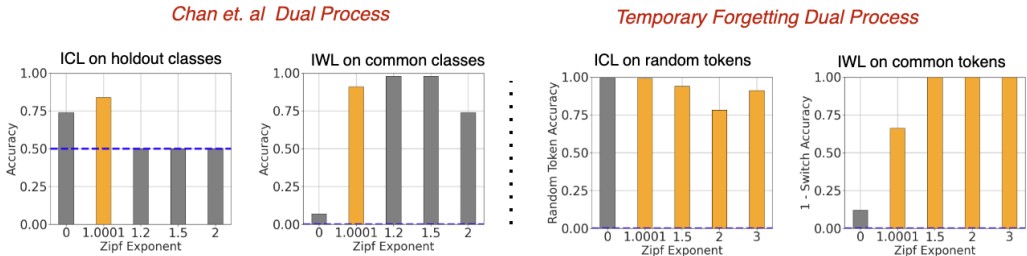

Figure 14: Temporary forgetting's ability to invoke dual processes (in yellow) on various distributions of our synthetic POS task compared with Chan et al. (2022b) observational baseline. Structural ICL and IWL are able to be co-occur in networks now trained on data distributions of any skew with $\alpha \geq 1$, as opposed to being limited to a specific "sweet spot" distribution.

## G   PUSHDOWN DATASETS

We use the train/dev splits from the English UD Treebank for the *c-pos*, *f-pos*, and *dep* tasks McDonald et al. (2013); the train/dev splits from Ontonotes-v5 in the CoNLL-2012 Shared Task format for the *ner*, *phrase start*, and *phrase end* tasks Linguistic Data Consortium (2013); Pradhan et al. (2012); the train/dev splits from Penn Treebank-3 for the *depth* and *dist* tasks Marcus et al. (1993); and generated token sequences for the *prev*, *dup*, and *ind* tasks.

We reproduce baselines from Elazar et al. (2020) to verify the correctness of our probing setups for *c-pos, f-pos, ner, dep, phrase start* and *phrase end* and from Hewitt & Manning (2019) for *depth* and *dist*.

## H   PUSHDOWN SIGNATURE OBSERVATION IN SYNTAX

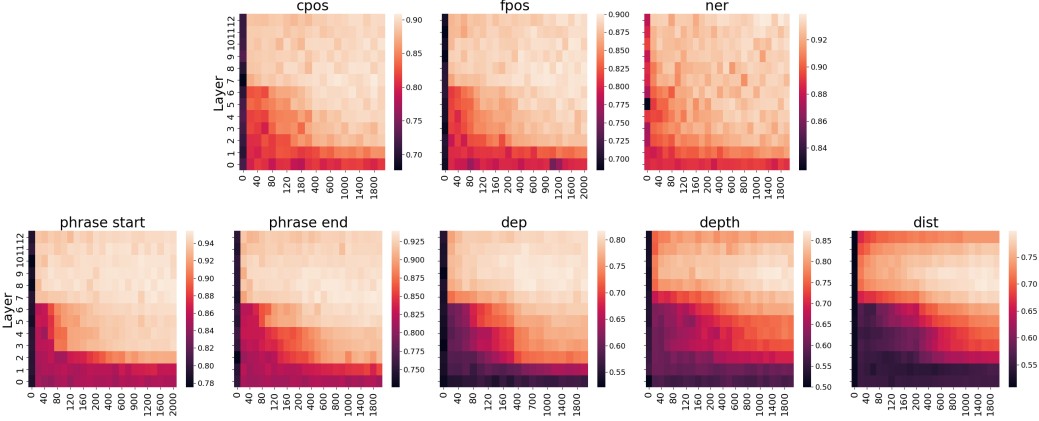

Figure 15: The "Pushdown Phenomenon" is observed across syntactic features, suggesting that a transition from IC to IW strategies happens across these features. In early steps of training, representing syntactic information occurs in later layers, which are more contextualized. However, as training progress, the same properties are better encoded in earlier layers due to memorization of token-level and n-gram level information. The n-gram level information requires attention to build, which explains why performance in *dep, depth*, and *dist* does not propagate all the way to embeddings.

We observe a "Pushdown Phenomenon", which describes a phenomenon where in early steps of training, computing token-wise syntactic properties occurs in later layers, which have more in-context information. However, as training progresses, the same properties are better encoded in earlier layers until only the first couple layers are required for representing syntactic properties.

We examine whether the "Pushdown Phenomenon" exists in various syntactic properties in BERT. To do so, we employ our probing setup (Appendix A) for the tasks of named entity recognition (*ner*), coarse part of speech (*c-pos*), fine-grained part of speech (*f-pos*), dependency parsing (*dep*), syntactic constituency boundaries which indicate the start and end of a phrase (*phrase start, phrase end*), depth in the parse tree (*depth*), and distance in the parse tree (*dist*). We probe each property across the axes of (1) training time steps and (2) layers. We repeat this process for three seeds of the MultiBERTs (Sellam et al., 2021). For all tasks, we probed all layers of MultiBERT seeds 0, 1, and 2 for timesteps from 0 to 200,000 increasing by 20,000; 200,000 to 1,000,000 increasing by 100,000; and 1,000,000 to 2,000,000 increasing by 200,000. If a specific word is composed of multiple subword tokens, we follow Hewitt & Manning (2019) and average the encoding across tokens.

We observe the "Pushdown Phenomenon" in all our examined tasks. However, we find that across tasks, syntactic information is "pushed down" at different rates. Early layer accuracy increases approximately follow a pattern of *ner* $\rightarrow$ *phrase start* $\rightarrow$ *cpos/fpos* $\rightarrow$ *phrase end* $\rightarrow$ *dep* $\rightarrow$ *depth* $\rightarrow$ *dist*. We leave it to future work to explore whether this timing is a function of (1) complexity of high-achieving rules/heuristics consistent with Belrose et al. (2024) or (2) a naturally occurring dependency hierarchy of syntactic relationships suggestive of implicit curriculum learning. One possible intuition for why the "Pushdown Signature" of memorization often coincides with poor maintenance of in-context strategies might be neural collapse (Parker et al., 2023; Rangamani et al., 2023), although this should be further investigated by future studies.

## I  SYNTHETIC POS TASK EXAMPLES

Here, we provide further details regarding the design of our synthetic POS task. Our task is designed to 1) minimally emulate a subtask performed in language models (Part-of-Speech tagging) while 2) controlling for various confounds. In particular (1) it does not allows heuristics based on token position and (2) is not deterministic based on the query.
Here are a couple clarifying examples (`<sequence>    <query>` $\rightarrow$ `<pattern>`):

1. (a) `is happy dog    dog` $\rightarrow$ `happy dog dog`
   (b) `dog is happy    dog` $\rightarrow$ `happy dog dog`
   Note that in this example, we show that using two templates rules out a simple position-based heuristic. If a model assumes that the noun occupies the 3rd position of the sequence, then the model will believe `happy` is a noun in the second example and falsely predict a response pattern of `dog dog dog`.

2. (a) `dog is happy    dog` $\rightarrow$ `happy dog dog`
   (b) `dog is sad    dog` $\rightarrow$ `sad dog dog`
   Note that in this example, both queries are `dog`, yet the predicted pattern is different. Context is necessary for correct prediction.

## J  TOY MODEL

We employ a 6-layer BERT model across the synthetic setting experiments. Experiments were performed with an MLM as syntactic structure is much more difficult to infer in autoregressive models as they are only exposed to an ordered subset of the tokens in a sentence. This model has 1 attention head per layer, 64-dimensional hidden dimensions, 128-dimensional intermediate representations, and tied weights for the embedding and unembedding layers. We optimize model parameters with AdamW with a learning rate of $5 \times 10^{-5}$ (Loshchilov & Hutter, 2019). The hidden dimension sizes were decided per a minimax strategy, i.e. this representation dimensionality was the smallest such that we achieved near perfect accuracy on a validation set for the downstream task. Future work should better examine the effect of representation size on in-context vs. in-weights learning.

# K    PERFORMANCE BY TOKEN DECILE

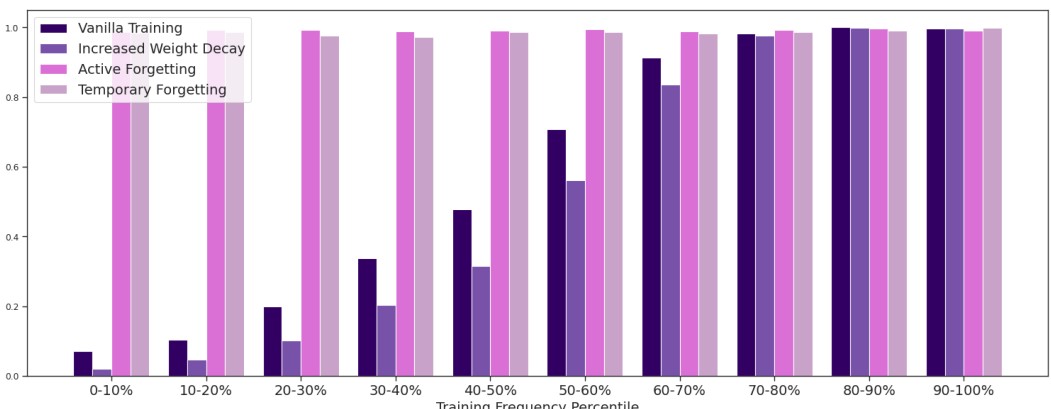

Figure 16: Increased weight decay has little/no effect on the failure of the structural ICL strategy (we increase weight decay from 0.01 to 0.1). In contrast, active and temporary forgetting boosts rare token validation accuracy significantly, as seen in the tail of the distribution. Parameters are $v = 10000, \varepsilon = 0.10, \alpha = 1.5$

We find that on highly skewed distributions, the tail of the distribution suffers immensely due to undertraining. This phenomenon cannot be rectified by Singh et al. (2023)'s method of promoting asymptotic ICL. However, we find that both active forgetting and temporary forgetting correct this behavior to boost performance on tail tokens in skewed distributions from near-zero to near-perfect levels.

## L  AMBIGUITY ($\varepsilon$) EXPERIMENTS

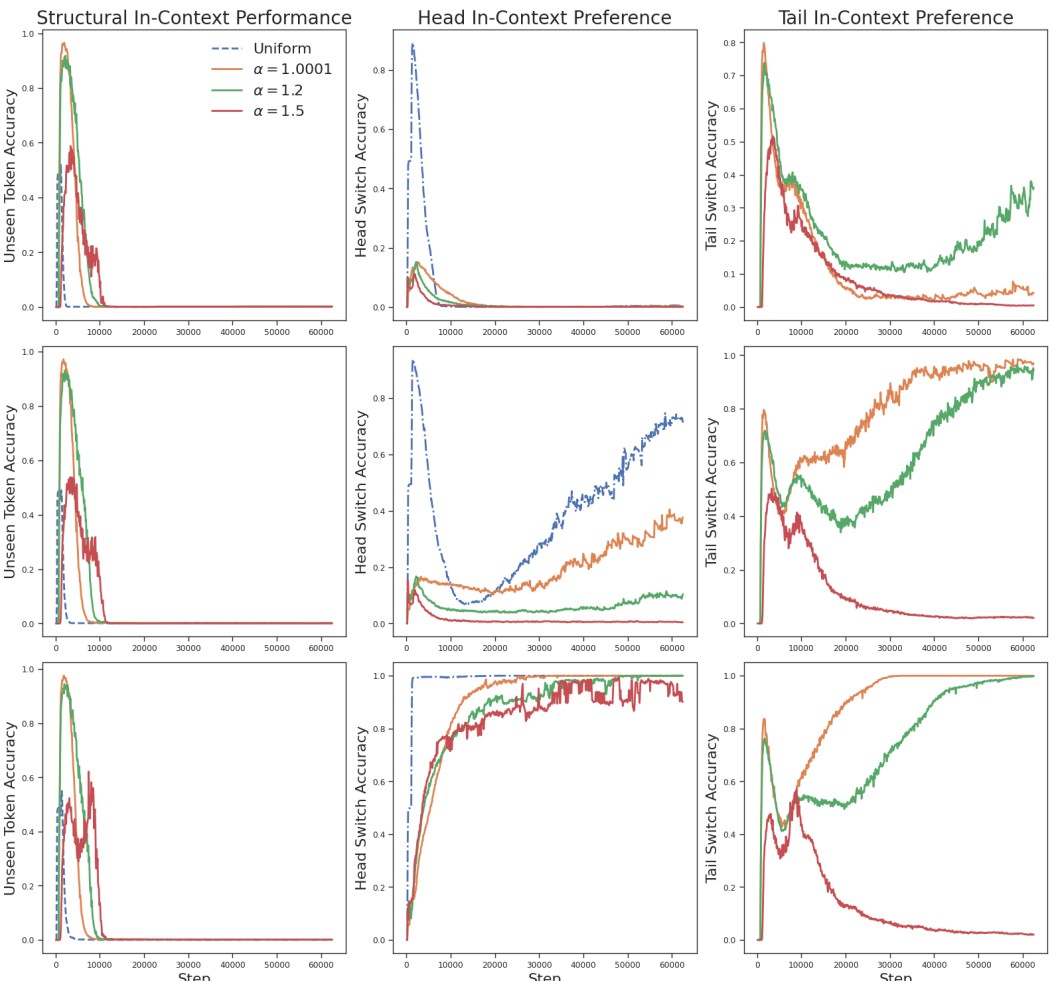

Figure 17: (Top) $\varepsilon = 0.01$, (Middle) $\varepsilon = 0.10$, (Bottom) $\varepsilon = 0.50$. Overall in-context strategy is dependent by amount of ambiguity in the labels. With 50% of the tokens as ambiguous, all unambiguous tokens use an in-context strategy; with 10%, there is a mixed strategy dependent on where in the distribution the example is; with 1%, almost unambiguous tokens use a memorized strategy. The vocab size is $v = 10000$.

In all of our ambiguity experiments, structural ICL is transient (even when 50% of tokens are ambiguous). The ambiguity parameter significantly alters the model's overall strategy. With a low ambiguity parameter, the model prefers memorization (IWL strategy) of unambiguous tokens and with a high ambiguity parameter, the model prefers an ICL strategy. Across all ambiguity parameters, there is a difference in tail and head behavior.

# M  VOCABULARY SIZE ($v$) EXPERIMENTS

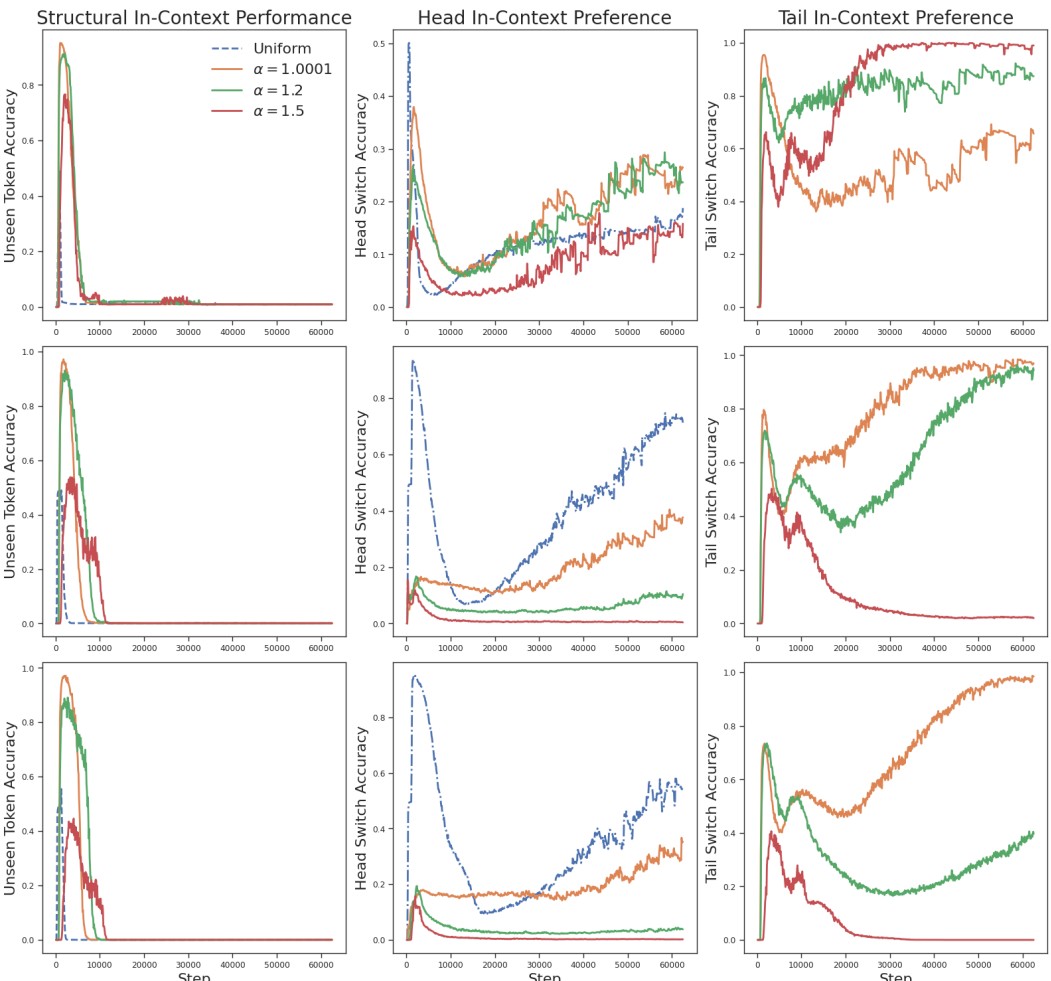

Figure 18: (Top) $v = 1000$, (Middle) $v = 10000$, (Bottom) $v = 20000$. The strength of an in-context solution depends on the interaction between vocabulary size $v$ and skewedness of the distribution $\alpha$. Too small of a vocabulary size (i.e. $v = 1000$) encourages more memorization in general but fixes performance in $\alpha = 1.5$ setting. The ambiguity is $\varepsilon = 0.10$.

In all of our vocabulary experiments, structural ICL is transient. As expected, we find that vocabulary size has a similar effect to the skewedness of the distribution. That is, increasing the vocabulary without bound would lead to poor tail ICL performance. Too small of a vocabulary size seems to increase ICL among very skewed distributions but decrease ICL among all other distributions.

## N    EMBEDDING ANALYSIS

We perform qualitative analyses on the embeddings produced by vanilla training (i.e. standard training without modification), active forgetting, and temporary forgetting in order to better understand how these training regimens impact model representations. These analyses, consisting of principal component analysis (PCA) and probing for POS, are located in Appendix N.

After vanilla training, the learned embeddings cluster according to their POS, far from the distribution of randomly-initialized tokens. We train a linear probe on these learned embeddings, and find that it can almost perfectly partition nouns and adjectives. Note that the disappearance of structural ICL occurs at the same time as the probe achieves above-random POS probing (i.e. memorization).

As expected, we do not see any structure in the embeddings produced after active forgetting. As such, a linear POS probe trained on these embeddings never achieves above random chance throughout training. The embedding distribution looks quite similar to the random initialization distribution, indicating that no information has been encoded in these embeddings. See Figure 19.

Finally, the temporary forgetting setting reflects aspects of both vanilla training and active forgetting; that is, the head of the token distribution learns to partition nouns and adjectives whereas the tail of the distribution does not learn any structure. The tail embeddings much more closely resemble the initialization distribution with temporary forgetting than with vanilla training. This results in a unseen token generalization in addition to memorized information. See Figure 20.

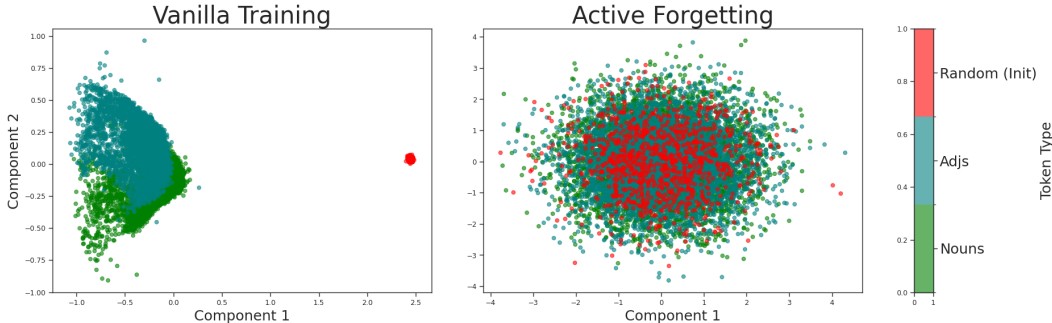

Figure 19: Vanilla training imposes structure on the adjectives and nouns such that randomly initialized (unseen) tokens are out-of-distribution whereas active forgetting embeddings resemble the initial distribution. Parameters used are $v = 10000, \alpha = 1.0001, \varepsilon = 0.10$.

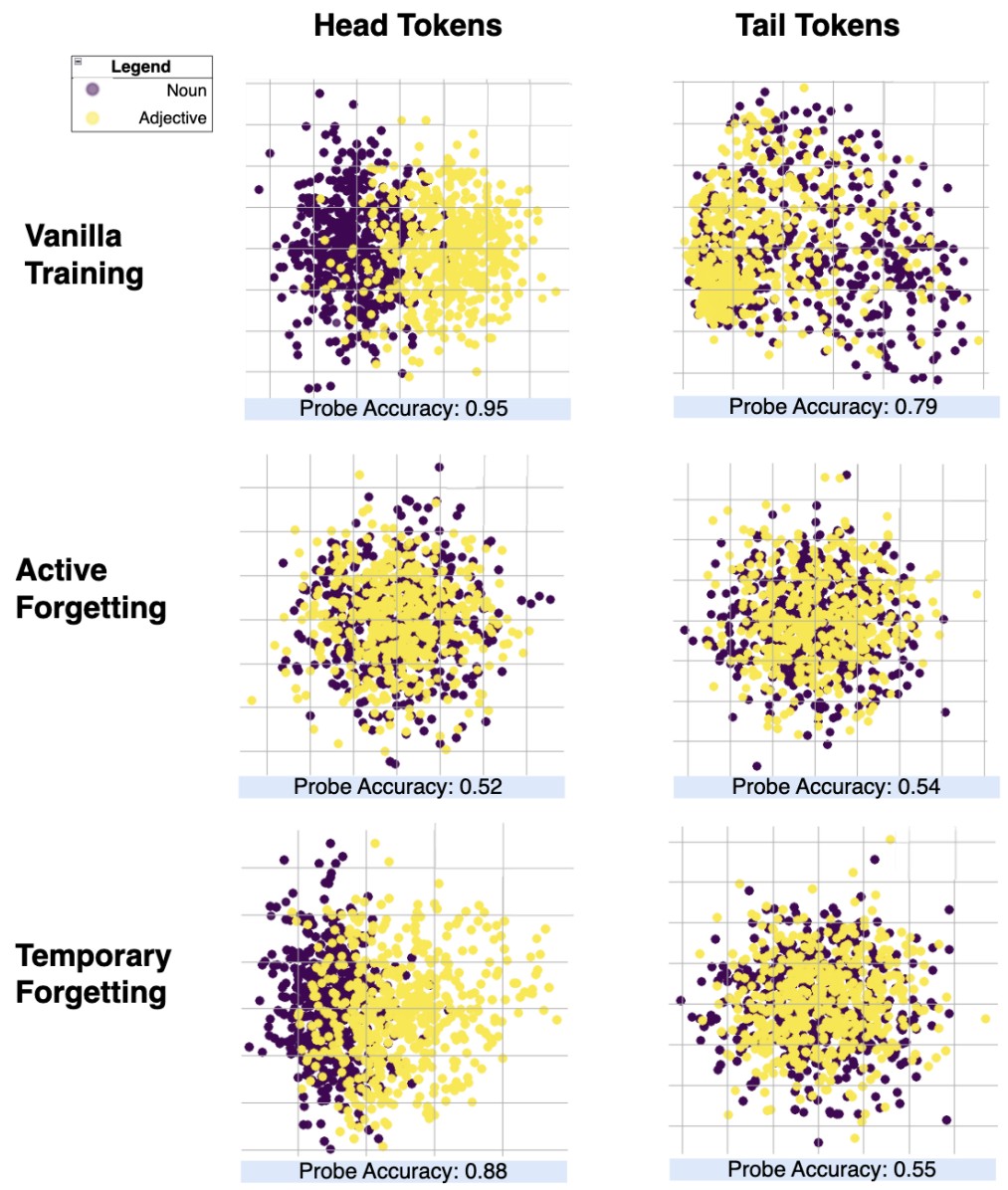

Figure 20: Vanilla training learns to partition noun and adjective embeddings in the head of the distribution, and some structure in the tail. Active forgetting learns no separation between noun and adjective embeddings. Temporary forgetting learns structure in the head of the distribution and no structure in the tail of the distribution. Parameters used are $v = 10000, \alpha = 1.2, \varepsilon = 0.10$.

## O   OTHER RANDOM DISTRIBUTION GENERALIZATION

Note that while we define structural in-context learning as free from reliance on any *encoded semantic information*, it is important to note that this does not mean that structural in-context learning assumes *no* geometry of the space. In fact, this would be practically impossible to achieve because connectionist networks function in a geometric space and take advantage of orthogonality, translation, scaling, etc. If we cannot make assumptions about the distribution from which the data is sampled, then we deprive our networks of their toolbox. Still, we test on random sampling distributions for the embeddings other than our initialization distribution. Namely, we test on a uniform distribution from 0 to 1 and a large normal distribution with mean of 5 and standard deviation of 5.

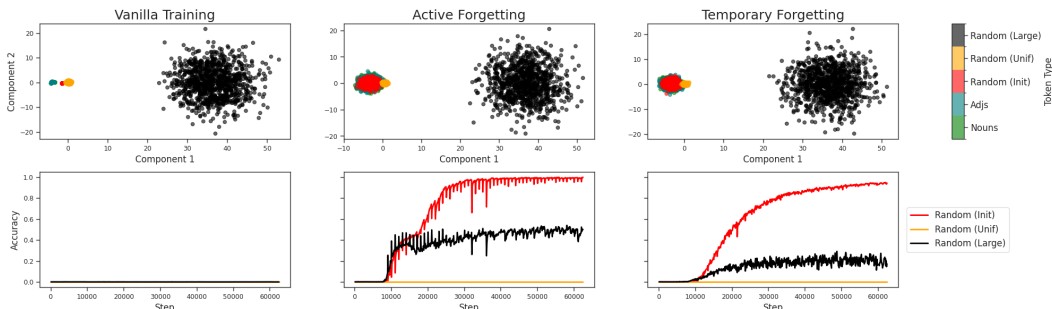

Figure 21: Vanilla training fails on all random tokens, whereas active/temporary forgetting succeed on the random distribution of initialization. Active and stop forgetting do not generalize to arbitrary random distributions, although show some generalization to normal distributions with large means and variances.

## P   REQUIRED COMPUTE FOR EXPERIMENTS

We employed compute resources at a large academic institution. We scheduled jobs with SLURM. For our naturalistic experiments, each MultiBERT seed required 24 separate runs (one per tested checkpoint at a particular timestep), which totaled $\approx$ 100 hours on an RTX A5000 with 24 GB of GPU memory. Over 3 seeds, this was $\approx$ 300 hours of GPU usage. For our synthetic setting, the vanilla training required 64 separate runs (one per hyperparameter combination of vocab size, ambiguity, and sampling distribution), which totaled $\approx$ 250 hours of RTX A5000 usage. Likewise, our active forgetting and temporary forgetting interventions took a similar amount of GPU usage. Therefore, in total, our GPU usage for all synthetic experiments summed up to about 750 hours. We ran experiments mostly in parallel with SLURM to iterate quickly. Compute was a significant limitation for the development time and informed our development of training interventions in a synthetic setting. In total, our GPU usage was significantly higher than the reported number due to various failed/modified experiments. The total compute likely was around 20,000 GPU-hours on RTX A5000s, although this is a rough estimate.

