# OpenReview forum: "Dual Process Learning: Controlling Use of In-Context vs. In-Weights Strategies with Weight Forgetting"
_ICLR.cc/2025/Conference — ICLR 2025 Poster_

### Official Review · Reviewer_zYZG · 2024-10-28

**Soundness:** 2
**Presentation:** 3
**Contribution:** 3
**Rating:** 6
**Confidence:** 5

**Summary:**

This paper studies the limitations and corresponding solutions for in-context learning with unseen tokens (i.e., structural ICL). Under this scenario, language models (LMs) learn the sentence structure (i.e., simple syntax structure) instead of token semantic meanings. Thus, with carefully designed forgetting strategies, the authors show that they can let LMs perform well under the structural ICL setting.

**Strengths:**

1.  This paper investigates an interesting research question: comparing ICL with parameter-tuning. Some explorations are helpful and provide useful insights.
2.  Besides, the idea of combining ICL with structures is inspiring. Most previous ICL works run experiments on the sentence classification datasets, which are highly semantic-related. The discussion about sentence structures instead of prompt structures are limited.

**Weaknesses:**

1. Some statements about ICL explanation are not up-to-date. The authors mentioned that typical ICL algorithms do not have stable learning manner and named it as conditional ICL. However, some recent works [1,2] have shown that ICL learns the task composition: the model only learns how to compose the task based on the learned tasks from pretraining, instead of learning from scratch based on the ICL examples. Thus, some claims in paragraph 2 of the introduction should be updated.

2. Experiment settings are a bit problematic. This work is based on previous findings. However, these findings are not general across all settings. For example, Singh et al., 2023 find that ICL slowly dissipates as models are overtrained ONLY for decoder-based LMs without pretraining. In their limitation section, they also mentioned that the conclusion may not hold for pretrained LMs (e.g., Multi-BERT here) under other scenarios. I can’t list all of them, but in general, what I mean is that we should be careful about the assumptions of previous work, and cannot simply generalize and rely on previous conclusions.

    Second, since this paper uses encoder-based LMs, it may not be able to perform ICL as there is little discussion about it. Besides, ICL ability is often used for large models. Implementing experiments on Multi-BERT with probing accuracy instead of generation is not convincing to show the conclusions of this paper is useful in practice.

3. The sentence structure is learned from pretraining instead of predefined. Thus, the model may not treat sentences as human do (i.e., with linguistic structures). Even if for the simple POS triple structure it doesn’t matter, other more general or complex structures may not work.

4. The potential impact of this work may be a bit limited. The motivation of injecting novel tokens under the ICL setting is not clear to me. As aforementioned, ICL is often used for large models, where their tokenizer vocabulary is large enough and it’s not necessary to add extra new tokens (e.g., LLaMA 3-8B). While for small models that may require to extend the tokenizer (frequently), it may not perform ICL well and they are not expected to be used under this scenario.

[1] Hahn M,; Goyal N. A theory of emergent in-context learning as implicit structure induction[J]. arXiv preprint arXiv:2303.07971, 2023.

[2] Li, J.; Hou, Y.; Sachan, M.; and Cotterell, R. What Do Language Models Learn in Context? The Structured Task Hypothesis. ACL 2024

**Questions:**

See weakness

---

> ### Author Response · Authors · 2024-11-19
>
> We thank you for your thoughtful review, and appreciate that you found the research question interesting!
>
> **W1 ("Some statements about")**: Thank you for the references! We have added these to lines 46-47 and believe they better contextualize our results and our experimental setting to the broader ICL literature. Let us know if there is anything else that might require updating.
>
> **W2 ("Experiment settings are")**: We agree, it is important to ensure that the assumptions of the previous work holds for our set of experiments. For the MultiBERT settings (encoder-only), we found the same trend as Singh et al., 2023 that the preference for ICL slowly dissipates which is shown in Figure 2 (Right). We have revised our draft to be more precise about our language regarding this point on line 151 and line 216.
>
> In addition, we replicated our set of active and temporary forgetting experiments on a decoder-only LM (GPT-2) with a task very similar to Chan et al., 2022 and Singh et al., 2023 to show that our results hold across model architectures.
>
> Finally, we have run another set of experiments on Pythia 1.4B (a decoder-only model) that also show structural ICL transience for generation which we describe in the main response! Hopefully, our additional set of experiments is more convincing to show that the conclusions are useful in practice!
>
> **W3 ("The sentence structure is ")**: We concur with your characterization of sentence structure learning. The model likely doesn’t treat sentences like humans do. However, prior work (Tenney et al., 2019; Clark et al., 2019) suggests that these models do strongly encode syntactic structure, which was the motivation for us using this particular model-task combination.
>
> In our research, we aim to close the gap between what a model can do with a familiar token in context compared to an unfamiliar token in context. Currently, rare ‘glitch’ tokens cannot perform simple tasks such as being repeated to the user. In a new set of experiments we describe in the main response and Appendix A.3.2, we tried another structure (a syllogism) in GPT-2 and found the same trend of structural ICL transience. We believe that the question of extending models to capture even more complex structures merits attention and requires additional experimentation to answer.
> Please let us know if this answers your question or if you have any follow up questions!
>
>
> **W4 ("The potential impact ")**: We look at the impact of our work as (1) discovering the issue of structural ICL transience and (2) proposing a solution that works in controlled and natural settings. We came up with a few natural examples that show the importance of robustness to undertrained tokens, namely (1) abstract reasoning tasks, (2) low-resource languages, and (3) code generation. We describe how undertrained tokens affect these tasks and how structural ICL would rectify these issues in the main response.
>
> Additionally, we have run another set of examples that show that on a natural language model (GPT-2), a probabilistic variant of temporary forgetting helps improve in-context strategies with unseen tokens. We describe them in additional detail in the main response and Appendix A.3.2. We hope that this convincingly shows that this technique is useful in natural language models for downstream applications.
>
> Thank you once again for your detailed comments and feedback. We hope that we’ve managed to address your concerns. If not, we’d be more than happy to continue the discussion!

---

> > ### Comment · Reviewer_zYZG · 2024-11-21
> > **Reply to the author reponse**
> >
> > Thanks for the explanation and new results!
> >
> > Including GPT-2 in the experiment is important to support some claims. Honestly, I'm not very satisfied with the responses to weakness 2 and 3. But consider that it's hard to provide very solid answers with one single paper, and my main concerns can be addressed more or less by the new findings. I'll change my decision to weak accept.
> >
> > Hope the authors could add the new results and improve the presentation later.
> >
> > Best,
> > Reviewer zYZG

---

### Official Review · Reviewer_E33F · 2024-11-02

**Soundness:** 4
**Presentation:** 4
**Contribution:** 2
**Rating:** 6
**Confidence:** 3

**Summary:**

The paper introduces a distinction between *structural* and *conditional* in-context learning, where structural ICL can generalise to novel tokens because it does not depend on token embeddings. They show that by default, structural ICL emerges and then sharply diminishes during pretraining, and they introduce a novel pretraining method based on periodically reinitialising the embedding matrix, which prevents the loss of structural ICL.

**Strengths:**

- The central thesis is clear
- The contributions are well-grounded in prior literature
- The distinction between structural and conditional ICL is compelling
- The experimental results support the thesis well, and are quite extensive, especially on the dynamics of structural versus conditional ICL

**Weaknesses:**

I feel there are two main weaknesses.

Firstly, it is not clear to me how important structural ICL actually is as an ability for models
  - The connection to glitch tokens is fair, but I don't expect these to be very common, and even when they do occur I'm not sure that preserving structural ICL is the best response - for instance, it wouldn't have fixed all the canonical solidgoldmagikarp problems, because structural ICL only preserves syntactic knowledge of glitch tokens, but fundamentally cannot give semantic knowledge of unknown tokens
  - The application to very rare tokens is likewise fair but by definition not very common, and again it is unclear to me whether in these cases there are any important capabilities which are purely syntactic

Secondly, it is not clear to me how far active/temporary forgetting impairs other abilities in models
- My intuition is that repeatedly reinitialising the embedding matrix is encouraging structural ICL by damaging other kinds of learning
- While the temporary forgetting approach might ameliorate this, it is not clear to me by how much or in what way it does this


Overall, I find much of the paper compelling in its presentation and exploration of structural ICL as a phenomenon, but I would be even more enthusiastic about it if it were clearer to me:
- that structural ICL is an important property (which could be demonstrated by natural examples of its relevance)
- what the side-effects are of the proposed methods for preserving structural ICL

**Questions:**

**Q1**

To what extent does active forgetting impair other model capabilities, if at all? What about temporary forgetting? I'd be interested to hear hypotheses, and particularly any experimental results you have.

**Q2**

Do you believe the problem of glitch tokens will recur in future, and in such a case, do you think preservation of structural ICL would be a useful strategy?

**Q3**

Can you give an example (or ideally a few examples) of cases where models might have a hard time performing a natural task because they are incapable of performing structural ICL on a rare token?

---

I have provisionally put down a marginal acceptance, but would be happy to revise my score upwards if these questions are appropriately addressed.

---

> ### Author Response · Authors · 2024-11-19
>
> We thank you for your detailed review! We appreciate that you found the distinction between structural and conditional ICL compelling.
>
> **Q1 ("To what extent" )**: In our experiments, active forgetting completely impairs the model’s capability to retain encoded information in the embeddings. Thus, the model would not be able to memorize properties of a token (part-of-speech, number, identity, etc) even if the token frequently appears in pretraining. This would result in poor performance when in-context learning is not possible. Figure 3 (Bottom Middle) shows that in-context Learning is preferred over in-weights learning even for head tokens. Additionally, Figure 12 (Right) in the appendix shows that active forgetting results in zero accuracy for in-weights learning problems. For this reason, we developed temporary forgetting, which aims to be much more practical.
> Temporary forgetting slightly diminishes the model’s capability to retain encoded information, instead preferring in-context strategies. This is evident in Figure 4 (Middle) and Figure 6 (Right) where the temporary forgetting setting has a slightly lower preference for in-weights learning for head tokens. We believe that this is a practical solution because it enables in-weights learning for common tokens while evoking structural in-context learning for rare/unseen tokens.
>
> **Q2 ("Do you believe" )**: This is a great question! The problem of ‘glitch tokens’ has been noted across open-source models (including GPT2, Llama 2, Mistral 7B, StableLM2 12B, etc) by Land & Bartolo, 2024. Prior research by Land & Bartolo, 2024 has posited this phenomenon arises due to the disconnect between tokenization and model training, which is likely to continue with the current LM training regimen.
>
> However, we believe that this is more deeply related to the properties of language. Natural language has been shown to follow a Zipfian distribution, which is by definition long-tailed (typos and internet-lingo exacerbates this effect). As such, there will always be tail tokens that are much rarer than head tokens, and thus will be relatively undertrained/unseen.
> While we agree with your characterization that structural ICL cannot instill semantic knowledge into these tokens, it brings the ability to use these tokens. For instance, when GPT-3 davinci-instruct-beta was asked “Can you repeat back the string 'SpaceEngineers' to me please?,” it responded with “"S-I-N-G-U-R-I-D"\ns-i-n-g-u-a-r-d.” (Rumbelow and Watkins, 2023). Behaviors such as evasion, hallucinatory completions, insults, etc occurred when using ‘glitch tokens’ with the prompt “Please can you repeat back the string '<token string>' to me?”
>
> With structural ICL, the model should correctly perform ICL algorithms such as repeating the string regardless of token identity. Moreover, it should be able to derive useful properties based on the context, such as the part-of-speech, number, etc.
>
>
> **Q3 ("Can you give an example" )**: Certainly! In addition to our example above, we came up with a few natural examples where models have difficulty because they are incapable of performing structural ICL on a rare token, namely (1) abstract reasoning tasks, (2) low-resource languages, and (3) code generation. We describe how undertrained tokens affect these tasks and how structural ICL would rectify these issues in the main response. Please let us know if this helps explain the importance or whether additional examples might be necessary.
>
> Thank you once again for your feedback. We’d be more than happy to continue the discussion!

---

> ### Comment · Reviewer_E33F · 2024-11-22
>
> Hmm, I am somewhat but not entirely persuaded.
>
> **Q1**
> I'm specifically curious whether you have any sense of how much temporary forgetting affects actual practical performance, eg/ any known benchmark. What does 'values' correspond to in Figure 6 (right)? It seems to be an entirely synthetic task.
>
> **Q2**
> I agree that glitch tokens will keep emerging if there's a tokenization/training mismatch, but as far as I can tell, your solution *also* requires a training intervention, and it's not obvious that this is a more practical intervention than just aligning the tokenizer.
>
> Is the davinci-instruct-beta example definitely not a glitch token problem?
>
> My impression is that the fact that natural language has many rare parts of text shouldn't matter assuming the tokenizer is trained on something representative of the training data, because normal tokenization strategies (eg/ BPE) will only produce tokens for fairly common substrings.
>
> **Q3**
>
> I don't find these examples enormously convincing. In all cases, I suspect this partly relies on a poorly configured tokenizer, and IWL doesn't actually help that much. For example, it would allow a model to make some basic inferences about the syntax of low-resource languages, or strange variable names, but these seem kind of fringe.
>
> --
>
> Essentially, I would be most compelled by an argument (preferably with empirical evidence) that this intervention is more practical and overall better for performance than simply making sure to use the right tokenizer, along with examples of frontier models (gpt4o-level) struggling with low-resource languages or code generation because of an inability to do basic syntactic inference of the kind SICL would enable
>
> I would be somewhat persuaded by
> - Extremely rare tokens appearing in frontier models (GPT4o-level) and causing trouble
> - Frontier systems continuing to have tokenizer/model mismatches (i.e. glitch tokens)

---

> > ### Author Response · Authors · 2024-11-23
> >
> > Thank you for your reply and further questions! We will clarify and expand our discussion around all of these points in the main text for the camera-ready submission.
> >
> > **Q1**: In short, we do not have a great sense of how temporary forgetting might impact practical performance on downstream tasks, as this would require us to train two entire language models from scratch (one with and one without temporary forgetting). Our experiments here instead serve as proof-of-concept that temporary forgetting appears to induce structural generalization with fairly little cost to in-weights learning. Moreover, our new experiments using pythia and GPT models on logical syllogisms also suggest that fine-tuning using a probabilistic form of temporary forgetting does not incur performance deficits and does induce structural in context learning.
> >
> > Despite these suggestive results, we acknowledge your concern, and will include some results in the final manuscript to directly address them. Specifically, we will run a few standard benchmarks on GPT-2 before and after probabilistic temporary forgetting fine-tuning, and report the results in the appendix of the camera-ready manuscript.
> >
> >
> > **Q2**: It is true that temporary forgetting as presented in the original submission requires retraining the language model, similarly to how one would need to retrain the language model after aligning the tokenizer. However, it is currently unclear whether and how one might actually align the tokenizer, while we provide a clear proof-of-concept strategy for mitigating this problem. Moreover, our new experiments on syllogistic reasoning provide an even more compelling and practical case: one might induce structural generalization by simply fine-tuning using a probabilistic form of temporary forgetting, which can rectify model behavior without the need to retrain the language model.
> >
> > **Q3**: We totally agree with your assertion that “IWL doesn’t actually help that much”, especially for our code and abstract logical reasoning examples. One can view both of these cases as examples of abstract reasoning over tokens, where the content of the token ought not impact the model’s answer. For these cases, the model actually should disregard in-weights information, and instead should adopt a structural ICL answer. However, as shown by prior work, models regularly do incorporate in weights information, leading to a variety of errors. All of our interventions encourage models to encode and deploy structural ICL functions when they need to.
> >
> > **Final Thoughts:**
> >
> > *"Examples of frontier models (gpt4o-level) struggling with low-resource languages or code generation because of an inability to do basic syntactic inference of the kind SICL would enable":*
> >
> > * Please see Wen et al. for evaluations of GPT-3.5-Turbo (not GPT4o, but still quite a good model!) incurring a great deal of NameErrors, which would be mitigated by increasing structural ICL competence.
> >
> > *"Extremely rare tokens appearing in frontier models (GPT4o-level) and causing trouble
> > Frontier systems continuing to have tokenizer/model mismatches (i.e. glitch tokens)":*
> >
> > * Fair enough! Here is a paper describing glitch/undertrained tokens in GPT4o, with a focus on under-resourced languages: https://arxiv.org/pdf/2406.11214.
> > * In addition, here is a twitter thread describing another glitch token in GPT4o (which we verified today, 11/22): https://x.com/Yuchenj_UW/status/1809336955005727092
> >
> >
> > *"Normal tokenization strategies (eg/ BPE) will only produce tokens for fairly common substrings:"*
> > * This paper shows that BPE and other normal tokenization strategies still follow Zipf’s law, with behavior on tail tokens that is different from behavior on head tokens: https://arxiv.org/pdf/2211.11041.
> >
> >
> > Once again, we appreciate your feedback! Please let us know if you have any follow-up questions!

---

> > > ### Author Response · Authors · 2024-11-27
> > > **Follow Up**
> > >
> > > Thank you sincerely for your thoughtful feedback and valuable suggestions! We hope our responses have addressed your concerns. Should you have any further questions or need additional clarification, please don't hesitate to let us know.
> > >
> > > If you feel that our responses and revisions enhance the quality and clarity of the work, we would greatly appreciate your consideration in updating your score.

---

### Official Review · Reviewer_pXza · 2024-11-03

**Soundness:** 3
**Presentation:** 2
**Contribution:** 3
**Rating:** 6
**Confidence:** 4

**Summary:**

This paper examines "structural" in-context learning (ICL), a specific type of ICL where some tokens in the model’s prompt are randomly initialized. The motivation for studying structural ICL is based on prior observations that ICL behaves unexpectedly when prompts include tokens that are rarely encountered during training, i.e., undertrained tokens. This paper uses synthetic tasks to demonstrate that structural ICL develops early in training but diminishes as training progresses. The authors show that periodically reinitializing token embeddings during training (active forgetting) preserves structural ICL abilities, albeit at the expense of memorizing information in the model’s embeddings. The paper further demonstrates that both structural ICL and memorization are maintained if active forgetting is employed, but only for the first N steps of training. This procedure is referred to as “temporary forgetting.”

**Strengths:**

1. I appreciated that this work distinguishes structural ICL from ICL in general. This seems like a natural and interesting extension to prior work that studies in-weights versus in-context learning.
2. Despite the lack of experiments on downstream tasks, I found the breadth of the experiments satisfactory. I also recognize the creativity employed by the authors given their compute constraints.

**Weaknesses:**

1. No experiments on downstream tasks. While the motivation to address the issue of undertrained tokens is interesting, it is difficult to assess how training with temporary forgetting will affect language models in real-world applications. Given the author’s limited computational resources, they might consider a variant of temporary forgetting that can be employed as a fine-tuning step, e.g., fine-tune with active forgetting for $k$ steps before resetting the token embeddings to their original states, leaving only the model parameters changed.
2. Limited technical novelty. Unless I misunderstood the paper, temporary forgetting is simply active forgetting applied for $K$ steps. This is only a minor weakness in my view and is excusable if the authors could find a way to perform experiments on downstream tasks.
3. The presentation can be improved. For example, there are instances where the figure legends contain terms that are not defined. In Figure 6 (right), “Vanilla Forgetting” is mentioned. Figure 11 uses the term “Stop Forgetting,” which I believe is intended to mean “Temporary Forgetting.”

**Questions:**

1. What are the limitations of temporary forgetting? Is it strictly beneficial, or did the authors find that some model abilities deteriorated? Given that evaluation wasn’t performed on downstream tasks, what synthetic experiments can be conducted to evaluate the trade-offs of temporary forgetting?
2. Suggestion: Since no evaluation was performed on downstream tasks, this paper could be improved by expanding on why studying undertrained tokens is important (beyond what is already discussed in the introduction). I am familiar with this area, so the introduction was adequate for me, but adding such a section might make this paper more broadly accessible.

---

> ### Author Response · Authors · 2024-11-19
>
> We thank you for your thoughtful review! We try to address the questions and concerns below:
>
> **W1 ("No experiments on downstream tasks")**: We agree that experiments on downstream tasks would be valuable to understand performance with a real-world LM. Thus, we fine-tune a GPT-large model for 2000 steps on a probabilistic version of temporary forgetting and describe the results in the main response and Appendix A.3.3 of the revised paper. This experiment shows that temporary forgetting can transfer to real-world LMs on downstream tasks. Please let us know if you have additional questions.
>
> **W2 ("Limited technical novelty")**: Your understanding is correct! However, our probabilistic version of temporary forgetting which we run in our new experiment for assessing performance on a downstream task in a real LM (syllogisms in GPT-2) has additional technical contributions described in the main response and Appendix A.3.3.
>
> **W3 ("The presentation can be improved")**: Thank you so much for catching these! We have fixed these in the revised draft.
>
> **Q1 ("What are the limitations")**: The limitations of training with temporary forgetting are that (1) it sometimes requires longer to converge (depending on the distribution skew) and (2) optimal parameters are not known a priori. These are described on lines 393-394 (for active forgetting, but also seen for temporary forgetting) and lines 530-531, respectively.
> We found that temporary forgetting did slightly diminish the model’s capability to retain encoded information while preserving structural ICL. This is evident in Figure 4 (Middle) and Figure 6 (Right) where the temporary forgetting setting has a slightly lower preference for in-weights learning for head tokens. This is a much more practical solution compared to active forgetting, which prevented the model from retaining information in its embeddings altogether.
> We evaluated trade-offs of temporary forgetting tests by assessing Tail, Head, Head Switch, Tail Switch, and Unseen Token Accuracy metrics for the synthetic task detailed in Section 4. Descriptions of these metrics can be found in Section 3.1.
> Our proof-of-concept shows structural ICL improvements in both decoder-only and encoder-only transformers on two different tasks. Additionally, we have now conducted an experiment which shows that a variant of temporary forgetting improves structural ICL on a downstream task in GPT-2 large (more details in main response and Appendix A.3.3).
>
> **Q2 ("Suggestion: Since no")**: This is a wonderful suggestion! We came up with a few natural examples that show the importance of robustness to undertrained tokens, namely (1) abstract reasoning tasks, (2) low-resource languages, and (3) code generation. We describe how undertrained tokens affect these tasks and how structural ICL would rectify these issues in the main response. Please let us know if this helps explain the importance or whether additional examples might be necessary.
>
> Let us know if you have any further questions, we would be happy to discuss!

---

> > ### Author Response · Authors · 2024-11-27
> > **Follow Up**
> >
> > We appreciate your feedback and the suggestions you’ve shared! We hope our responses have resolved any concerns you may have, but please don’t hesitate to reach out if you need further clarification or have additional questions.
> >
> > If you believe our responses and revisions have improved the clarity and overall quality of the work, we would be grateful if you might consider adjusting your score accordingly.

---

> > > ### Comment · Reviewer_pXza · 2024-12-02
> > >
> > > Thank you for the response and revisions. The presentation has been improved, and I appreciate the attempt to introduce additional technical novelty and demonstrate applicability to real-world LMs. However, I did not find the revisions compelling enough to raise my score above a 6.

---

### Official Review · Reviewer_Szdj · 2024-11-04

**Soundness:** 2
**Presentation:** 3
**Contribution:** 2
**Rating:** 6
**Confidence:** 3

**Summary:**

This paper studies the emergence and disappearance of in-context vs in-weights learning in masked language models; in particular, a kind of ICL termed “structural ICL”, which refers to the ability to infer (and then use) information about the structural role (e.g., POS) of words from the context. In experiments, this ability appears and then disappears over the course of MLM training. A recently proposed “active forgetting” approach helps keep the ability, leading to models that use both in-context and in-weights learning.

**Strengths:**

- The paper studies a question of substantial interest, nicely continuing prior work on different kinds of learning
- considers both real-world LM and toy models
- Provides novel insights by distinguishing "structural" from "conditional" ICL
- Provides method for mitigating the loss of structural ICL in a toy setup

**Weaknesses:**

- Structural ICL is operationalized using performance on random token embeddings. However, as the paper also demonstrates (Figure 18), such embeddings are drastically OOD for a trained model. Wouldn't a fairer test use novel embeddings that match the distributional properties of the embeddings of the trained model?
- Results on Structural ICL in MultiBERT (Section 3) are based on POS = noun vs adjective as the only property. It remains unclear how robust findings are to other properties, such as other pairs of POS, or other properties, such as grammatical number.
- Results on Structural ICL in MultiBERT (Section 3) are based only on probing accuracy, not behavior of the model. This seems suboptimal given that in-context learning as an emergent property of LLMs is generally thought of as an ability appearing in next-token prediction (e.g., complete a prompt with an appropriate label), One could test for ICL in properties such as POS behaviorally by creating contrastive examples where the key word has different POS and comparing model probabilities.
- The term “Dual Process” here is used to refer to models using both in-weights and in-context learning, and the paper refers to Kahneman 2011; Miller 2000. The link to dual process theory appears tenuous: dual process theory refers to thse use of implicit, unconscious vs explicit, conscious processes; the link to in-weights vs in-context inference is at best hand-waved in line 90-91, but no explicit justification for the link is given. The choice of the term and the link to dual process theory thus appears a bit of a stretch to apply the in this context.
- It remains unclear if the proposed strategy for keeping structural ICL abilities would transfer to real-world LMs. While training an LLM is understandably out of scope, even a modest LM could provide substantial insight here.

**Questions:**

See weaknesses.

Minor note: Line 317-8: it seems the sentence is missing a verb

---

> ### Author Response · Authors · 2024-11-19
>
> We appreciate the thoughtful feedback, and are glad you thought that our question was of substantial interest!
>
> **W1 ("Structural ICL is operationalized")**: This is a great question! We define structural ICL as in-context learning where one or more tokens do not contain encoded information (i.e. without the in-weight signal of the word embedding). This enables us to study unseen and undertrained tokens and rectify behavior. For this reason, we sample novel embeddings drawn from the initialization distribution for the embedding matrix. We are guaranteed that these initialized embeddings have not learned any information via backprop.
> In contrast, if we constructed novel embeddings that match the distributional properties of the embeddings via something like linear interpolation of known embeddings or manifold sampling, then our embeddings would likely still possess encoded information that represent some weighting of the known embeddings we derive it from (similar to how sentences are often represented as the mean of token embeddings).
>
> **W2 ("Results on Structural ICL")**: This was a recurring comment that reviewers had! We understand the concern to generalizability on language models and thus have run an additional experiment that expands the results about Structural ICL to Pythia 1.4B (a decoder-only transformer model) with a syllogism-based test. We detail this experiment in our main response and in Appendix A.3.2 of the revised paper. Furthermore, we have experiments on encoder-only and decoder-only synthetic models that also show the same trend of Structural ICL transience in Section 4.1 and 7.
>
> **W3 ("Results on Structural ICL ")**: In designing the MultiBERT experiment, we chose probing because (1) we wanted to analyze model strategies early in training when generation would likely be poor and (2) we were interested in learning how model strategies were represented internally. However, we agree that there is value in showing this phenomenon behaviorally, and thus have conducted an additional experiment reproducing this result in Pythia models (decoder-only transformer model) with a syllogism-based generation test. We hope this experiment satisfies concerns!
>
> **W4 ("The term “Dual Process” here ")**: We refer to the framework of dual processes described in Moskovitz 2022, a cross-disciplinary framework where "two mechanisms appear to operate concurrently, one relatively high in computational complexity, the other relatively simple." We believe that the mechanisms of in-weights/in-context inference loosely fits into this dual-process framework, as in-weights strategies are relatively simple compared to in-context learning and that both systems compete for use. Still, you are correct that this is based on only a loose rationale. We are borrowing terminology to succinctly describe the phenomenon rather than making concrete, evidence-based parallels. We have added a footnote on page 2 to not overstate the similarity to other dual processes and have added the Moskovitz 2022 citation.
>
> **W5 ("It remains unclear if the ")**: As temporary forgetting is newly proposed, we understand the utility in testing on a real-world LM. Thus, we fine-tune a GPT-2 large model for 2000 steps on a probabilistic variant of temporary forgetting and describe the results in the main response and Appendix A.3.3 of the revised paper. This experiment shows that our proposed strategy can transfer to real-world LMs on downstream tasks. Please let us know if you have additional questions.
>
> Please let us know if you have any further questions, we would be happy to discuss!

---

> > ### Comment · Reviewer_Szdj · 2024-11-21
> >
> > Thank you for the thoughtful response. I will read the revised draft in detail as soon as I can.

---

> > > ### Comment · Reviewer_Szdj · 2024-11-24
> > >
> > > I believe the paper has become stronger, and I will update my score accordingly.

---

### Author Response · Authors · 2024-11-19
**Main Response**

# Main Response:

We thank the reviewers for their detailed feedback and questions. Their comments and suggestions have helped us to make significant improvements to the work! Below are responses to common questions.

## Motivating Examples/Importance

Below, we provide a few instances where models have difficulty because they cannot perform structural ICL on a rare token. These are mainly symbolic reasoning tasks, which are important for widespread use cases, including those enumerated below.
1. **Abstract reasoning tasks:**
Dasgupta et al., 2022 evaluate large language models on abstract reasoning tasks and find many imperfections, which according to them “is frequently highlighted as a crucial missing component of current AI.” For instance, they test Natural Language Inference (NLI) on nonsense words (e.g. If vuffs are bigger than feps, then feps are smaller than vuffs) and find that models cannot accurately predict statement validity above chance (Dasgupta et al., 2022 - Figure 3). While nonsense words are not natural, we would often want to introduce words unfamiliar to the model such as names, places, events, etc and would like to preserve logical reasoning. Moreover, it is highly likely that additional words become a part of the common lexicon as language is continually evolving while the training data has a static cutoff.

2. **Low-resource languages:**
Large language models perform poorly on in-context learning tasks involving low-resource languages, such as Swahili. Various works explore X-ICL (cross-lingual in-context learning) methods, which incorporate source language exemplars into context (Winata et al., 2021b; Shi et al., 2023). However, these still fail to compete with a simple translate-test, which translates the query and then conducts zero-shot inference in a high-resource language (Cahyawijaya et al., 2024). These languages have much difficulty as tokens are quite rare in pretraining. Structural ICL would enable much higher ICL performance.

3. **Code Generation:**
Models are often used for code generation, where function and variable names are often rare or unique (but properties should be derivable based on the context). Wen et al., 2024 analyzed 14 LLMs and found ~22% of code generation errors attributable to NameErrors. Structural ICL would ensure robustness to naming in code generation, which would likely mitigate many of these errors.

We have modified the manuscript to briefly mention these examples in lines 516-517 and will expand on these examples in the camera-ready version.


## Reproducing Structural ICL Transience on Generation Task

Multiple reviewers had concerns about the generalizability of our MultiBERT experiments, asking us to examine behavior. We designed another task to do just this:

Our task requires abstract reasoning on untrained tokens in a decoder-only transformer. We give the model the syllogism template

<X> is <Y>.

<Y> is <Z>.

Therefore, <X> is __

The correct answer is <Z>. We examine accuracy, which we find by looking at the probabilities of predicting <Z> versus <Y>. We test baseline performance over training steps where <X>, <Y>, <Z> are chosen from the set of tokens representing A-Z, and we test unseen token performance by replacing <X> with an unseen token. We test on checkpoints released from Pythia-1.4B (Biderman et al., 2023), a decoder-only language model, and reproduce the trend of Structural ICL emerging and then quickly disappearing (as seen by unseen token performance). Additional details about the setup and results are located in Appendix A.3 of the revised manuscript.


## Probabilistic Temporary Forgetting during Fine Tuning of GPT-2

Most reviewers deemed that it would be very valuable to test how temporary forgetting might work in a natural language model setting. We strongly agree, and thus designed an experiment on GPT-2 large, a decoder-only language model using the same syllogism task.

In our experiment, we finetune GPT-2 large (Radford et al., 2019) on wikitext sentences for 2000 steps. To accommodate the fine-tuning setting, we use a probabilistic variant of temporary forgetting: every step, we replace tokens in the batch with p=0.10 with randomly initialized embeddings. After the step, we set the embedding matrix back to its original values, hence maintaining the character of temporary forgetting.

We find that syllogism accuracy with unseen tokens jumps from 0.02 to 0.927 while the baseline syllogism accuracy remains the same. This jump does not occur with vanilla fine-tuning.

We believe that this experiment shows that temporary forgetting is useful in real language modeling settings on downstream tasks. Additional details are included in Appendix A.3.3, and we plan on polishing these experiments for the camera-ready version. Moreover, we plan to put our fine-tuning experiment in the main body of the paper. Please let us know if you have any questions!

Once again, we sincerely appreciate your feedback!

---

> ### Author Response · Authors · 2024-11-19
> **References**
>
> Biderman, S., Schoelkopf, H., Anthony, Q. G., Bradley, H., O’Brien, K., Hallahan, E., Khan, M. A., Purohit, S., Prashanth, U. S. V. S. N. S., Raff, E., & others. (2023). Pythia: A suite for analyzing large language models across training and scaling. In Proceedings of the International Conference on Machine Learning (pp. 2397–2430). PMLR.
>
> Dasgupta, I., Lampinen, A., Zoran, D., Santoro, A., & McClelland, J. L. (2022). Language models show human-like content effects in reasoning. Retrieved from https://web.stanford.edu/~jlmcc/papers/DasguptaLampinenEtAl22LMsShowHumanLikeContentEffectsInReasoning.pdf
>
> Cahyawijaya, S., Lovenia, H., & Fung, P. (2024). LLMs Are Few-Shot In-Context Low-Resource Language Learners. In Proceedings of the 2024 Conference of the North American Chapter of the Association for Computational Linguistics: Human Language Technologies (NAACL 2024). Association for Computational Linguistics.
>
> Wen, H., Zhu, Y., Liu, C., Ren, X., Du, W., & Yan, M. (2024). Fixing Code Generation Errors for Large Language Models. arXiv preprint. https://arxiv.org/pdf/2409.00676v1
>
> Clark, K., Khandelwal, U., Levy, O., & Manning, C. D. (2019). What Does BERT Look at? An Analysis of BERT's Attention. In Proceedings of the 2019 ACL Workshop BlackboxNLP: Analyzing and Interpreting Neural Networks for NLP (pp. 276-286). Association for Computational Linguistics.
>
> Tenney, I., Das, D., & Pavlick, E. (2019). BERT Rediscovers the Classical NLP Pipeline. In Proceedings of the 57th Annual Meeting of the Association for Computational Linguistics (pp. 4593-4601). Association for Computational Linguistics.
>
> Radford, A., Wu, J., Child, R., Luan, D., Amodei, D., & Sutskever, I. (2019). Language models are unsupervised multitask learners. OpenAI Blog, 1(8), 9. Retrieved from https://www.openai.com/research/language-models-are-unsupervised-multitask-learners
>
> Chan, S.C.Y., Santoro, A., Lampinen, A. Wang, J., Singh, A., Richemond, P.H., McClelland, J. & Hill F.. Data distributional properties drive emergent in-context learning in transformers, 2022b.
>
> Moskovitz, T., Miller, K., Sahani, M., & Botvinick, M.. A unified theory of dual-process control, 11 2022.

---

### Meta-Review · Area_Chair_MKuY · 2024-12-11

**Metareview:**

The authors continue the thread of existing work on ICL and token-sensitivity to define "structural in-context learning". They show that this has a different pattern of emergence than general ICL and suggest a pre-training method to improve it. Issues raised about applicability to different models and real-world relevance are discussed.

The paper is relevant and the experiments solid if not revolutionary, recommend accept as poster.

**Additional Comments On Reviewer Discussion:**

Several clarifications and additions (e.g. adding GPT2 results) were made during rebuttal that addressed reviewer concerns and resulted in score changes.

---

### Decision · Program_Chairs · 2025-01-22

Accept (Poster)